# Causal Disentangled Anchor Learning for Scalable Fair Multi-view Clustering

**Suyuan Liu** [1]  **Shengfei Wei** [1]  **Wenjing Yang** [1]  **Shengju Yu** [1]  **Siwei Wang** [2]  **Xueqiong Li** [1]  **Wenpeng Lu** [3 4]
**Xinwang Liu** [1]

## Abstract

Existing fair multi-view clustering methods typically suffer from a severe trade-off between clustering utility and fairness, while incurring prohibitive quadratic complexity on large-scale datasets. To address these challenges, we propose Causal Disentangled Anchor Learning (CDAL), a novel framework that achieves scalable fairness via structural disentanglement. Guided by a structural causal model perspective, CDAL utilizes a dual-anchor mechanism to structurally separate latent representations into orthogonal semantic and sensitive subspaces. We further ensure statistical independence through a linearized Hilbert-Schmidt Independence Criterion (HSIC) constraint, which is optimized via an efficient alternating scheme. Theoretically, we prove the identifiability of the disentangled factors and guarantee the algorithm's global convergence and linear time complexity $\mathcal{O}(n)$. Extensive experiments on large-scale benchmarks demonstrate that CDAL outperforms state-of-the-art methods, achieving a superior utility-fairness trade-off. Our code is publicly available at https://github.com/Tracesource/CDAL.

## 1. Introduction

Multi-view clustering (MVC) has garnered significant attention in the machine learning community due to its capability to integrate complementary information from diverse data

[1]College of Computer Science and Technology, National University of Defense Technology, Changsha, China [2]Academy of Military Sciences, Beijing, China [3]Key Laboratory of Computing Power Network and Information Security, Ministry of Education, Shandong Computer Science Center, Qilu University of Technology, Jinan, China [4]Shandong Provincial Key Laboratory of Computing Power Internet and Service Computing, Shandong Fundamental Research Center for Computer Science, Jinan, China. Correspondence to: Xueqiong Li <lixueqiong13@nudt.edu.cn>, Xinwang Liu <xinwangliu@nudt.edu.cn>.

*Proceedings of the 43rd International Conference on Machine Learning*, Seoul, South Korea. PMLR 306, 2026. Copyright 2026 by the author(s).

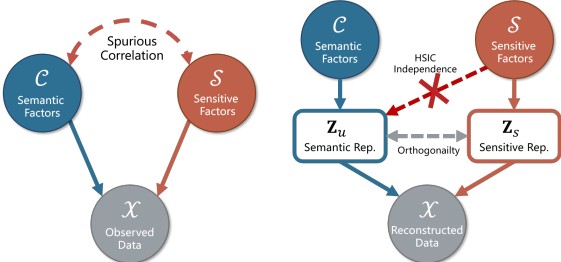

(a) Biased data generation process     (b) Causal disentanglement via CDAL

*Figure 1.* **Illustration of the causal disentanglement mechanism versus the entangled data generation process. (a)** The observed data $X$ is generated by latent semantic factors $C$ and sensitive factors $S$. A spurious correlation often exists between $C$ and $S$ due to historical biases. Traditional methods exploit this correlation, leading to biased outcomes. **(b)** Our framework simulates an intervention via structural constraints. It explicitly learns disentangled representations, $\mathbf{Z}_u$ (semantic) and $\mathbf{Z}_s$ (sensitive). The core mechanism severs the causal link between the sensitive factors $S$ and the semantic representation $\mathbf{Z}_u$ with the HSIC independence constraint, while orthogonality ensures subspace separation.

sources (Cui et al., 2025; Li et al., 2023; Fei et al., 2025; Liang et al., 2021; 2026). By fusing consensus patterns across views, MVC has demonstrated superior performance in various applications, such as computer vision and bioinformatics (Hu et al., 2024; Zhu et al., 2022; Lin et al., 2026; Li et al., 2026). However, as machine learning algorithms are increasingly deployed in high-stakes domains like credit scoring, recruitment, and medical diagnosis, the fairness of clustering results regarding sensitive attributes (e.g., gender, race) has become a critical ethical constraint (Shaham et al., 2025; Kleindessner et al., 2019). Biased clustering results can perpetuate or even exacerbate societal discrimination, necessitating the development of Fair Multi-view Clustering (FMVC) methods (Zhao et al., 2024; Xu et al., 2025; Wan et al., 2024b;a).

Despite recent advancements, existing FMVC approaches typically face substantial theoretical and practical challenges (Xu et al., 2025). A primary limitation lies in the fairness-utility trade-off. Most prevalent methods adopt a correlation suppression paradigm, which forces the learned representations to be statistically independent of sensitive attributes via aggressive regularization or adversarial train-

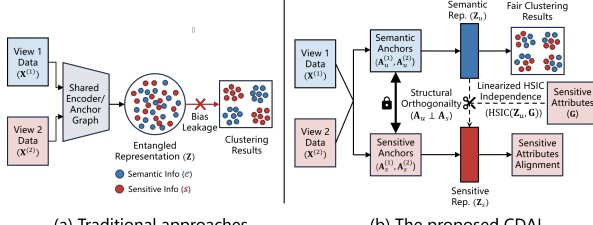

(a) Traditional approaches          (b) The proposed CDAL

*Figure 2.* **Comparison between traditional approaches and the proposed CDAL framework. (a)** Traditional approaches often learn entangled latent representations where semantic information (blue) and sensitive information (red) are mixed, leading to bias leakage in clustering results. **(b)** CDAL structurally disentangles these factors via a novel dual-anchor mechanism. It enforces physical separation through structural orthogonality between semantic anchors ($\mathbf{A}_u$) and sensitive anchors ($\mathbf{A}_s$), and statistical independence via a linearized HSIC constraint between the semantic representation ($\mathbf{Z}_u$) and sensitive attributes ($\mathbf{G}$), ensuring both fairness and clustering utility.

ing (Wang et al., 2025; Li et al., 2020). As illustrated in Figure 2(a), by treating data as a black box and ignoring the underlying generation mechanism, these methods often learn entangled representations where sensitive information inevitably leaks into clustering tasks (Park et al., 2021). They tend to indiscriminately eliminate information correlated with sensitive attributes, even when such information is essential for preserving clustering semantics, resulting in a significant degradation of clustering accuracy. Compounding this challenge is the prohibitive computational complexity. State-of-the-art fair clustering methods frequently rely on spectral decomposition or kernel-based independence measures, which inherently involve the construction and manipulation of $n \times n$ matrices (Tonin et al., 2025). The resulting $\mathcal{O}(n^2)$ or $\mathcal{O}(n^3)$ complexity renders them impractical for large-scale datasets, severely restricting their deployment in real-world scenarios.

To bridge these gaps, we rethink the problem from a structural causal model perspective. As depicted in Figure 1, we contrast the biased data generation process with our proposed intervention. We posit that the observed multi-view data is generated by two latent factors: a semantic factor $C$ (cluster-invariant) and a sensitive factor $S$ (bias-specific). In the real world (Figure 1(a)), these factors are often spuriously correlated. Traditional methods that directly learn from $X$ inevitably entangle these factors. In contrast, our framework (Figure 1(b)) aims to structurally isolate these factors at the source by imposing constraints derived from the assumed causal graph. Therefore, achieving a superior Pareto frontier between fairness and utility requires an explicit disentanglement mechanism capable of physically separating semantic content from sensitive bias, rather than merely suppressing their correlations in the output representation.

In this paper, we propose a novel framework termed Causal Disentangled Anchor Learning (CDAL), which achieves scalable and fair multi-view clustering via structural disentanglement, as illustrated in Figure 2(b). To address the scalability issue, we adopt an anchor-based strategy but redesign it with a dual-anchor mechanism. Specifically, we learn two distinct sets of anchors: semantic anchors spanning the cluster-relevant subspace and sensitive anchors spanning the bias-specific subspace. By imposing a structural orthogonality constraint between these two sets of anchors, we physically block the propagation of sensitive information into the semantic representation. Furthermore, to ensure statistical independence, we introduce a Linearized Hilbert-Schmidt Independence Criterion (HSIC) constraint. Unlike traditional kernel methods, our formulation leverages the low-rank property of anchor graphs to estimate independence in linear time $\mathcal{O}(n)$, making our method applicable to massive datasets.

The main contributions of this paper are summarized as follows:

- We propose CDAL, a novel framework that addresses the fairness-utility trade-off from a causal perspective. By structurally disentangling semantic and sensitive factors via a dual-anchor mechanism, CDAL effectively filters out bias while preserving cluster-discriminative structures.

- We derive a rigorous alternating optimization algorithm where each sub-problem admits a closed-form solution. Notably, by linearizing the HSIC constraint, we reduce the overall time complexity from quadratic to linear $\mathcal{O}(n)$, significantly outperforming existing fair clustering baselines in efficiency.

- Extensive experiments on large-scale benchmark datasets demonstrate that CDAL achieves a superior utility-fairness trade-off. Our method consistently outperforms state-of-the-art baselines, delivering significant improvements in fairness while maintaining high clustering accuracy.

## 2. Related Work

### 2.1. Fair Clustering

Fair clustering aims to mitigate systematic biases by ensuring balanced demographic proportions within clusters, a concept pioneered by Chierichetti et al. (2017). However, state-of-the-art methods typically rely on spectral decomposition or kernel-based measures involving $n \times n$ similarity matrices, incurring prohibitive computational costs for large-scale data (Chhabra et al., 2021). Even recent efficiency-focused approaches remain bottlenecked by expensive graph operations, such as eigen-decomposition (Wang et al., 2023)

or $k$-NN graph construction (Li et al., 2024). Furthermore, these methods often enforce fairness via indiscriminate correlation suppression, which inevitably compromises semantic integrity and clustering utility, highlighting the need for structural causal disentanglement.

### 2.2. Anchor Graph Learning

Anchor graph learning was introduced to address the scalability bottleneck of graph clustering by approximating the similarity matrix via a bipartite graph between samples and a small set of anchors (Liu et al., 2010; Yang et al., 2022; Yu et al., 2025). Recent advancements have shifted from static heuristics to adaptive mechanisms that capture complex topological structures. For instance, Feng et al. proposed a tensorial projection approach to map anchor graphs directly into the label space (Feng et al., 2025). Wang et al. introduced heterogeneous graph filtering to align view-specific structures efficiently (Wang et al., 2024). Despite reducing computational complexity to linear time, these methods indiscriminately treat all information encoded in anchors as beneficial for clustering. Consequently, they fail to filter out sensitive biases such as gender or race, leading to discriminatory outcomes. In the following section, we propose a dual-anchor mechanism that overcomes this limitation by structurally isolating sensitive factors from the semantic anchor space.

### 2.3. Disentangled Representation Learning

Disentangled representation learning aims to uncover independent latent factors of variation (Bengio et al., 2013). While extensively used in MVC to separate view-consistent from view-specific details (Lu et al., 2024; Liu et al., 2024), its application in FMVC shifts to distinguishing semantic variables from sensitive attributes (Jiang et al., 2025; Xu et al., 2025). However, most existing FMVC methods rely on implicit correlation suppression rather than explicit structural disentanglement, effectively treating data generation as a black box. In contrast, our CDAL framework enforces structural disentanglement via orthogonal anchor subspaces, physically blocking bias propagation at the graph construction level.

## 3. Methodology

In this section, we present the Causal Disentangled Anchor Learning (CDAL) framework. We first formalize the problem from a structural causal perspective. We then detail the proposed structural disentanglement mechanism and the linearized independence constraint. Finally, we provide the unified objective function, an efficient alternating optimization algorithm, and theoretical analysis regarding identifiability and complexity.

### 3.1. Problem Definition

Let $\mathcal{X} = \{\mathbf{X}^{(v)} \in \mathbb{R}^{d_v \times n}\}_{v=1}^{V}$ denote the multi-view data matrix consisting of $n$ samples across $V$ views, where $d_v$ is the feature dimension of the $v$-th view. Let $\mathbf{G} \in \{0,1\}^{h \times n}$ be the sensitive attribute matrix (e.g., gender, race) associated with the samples. Our primary objective is to partition the $n$ samples into $k$ distinct clusters. Ideally, the clustering assignments should accurately capture the intrinsic semantic structure of the data while remaining statistically independent of the sensitive attributes $\mathbf{G}$ to ensure fairness.

### 3.2. Structural Causal Disentanglement

To achieve the aforementioned goal, we assume the data generation process follows a specific structural causal model. Based on this prior knowledge, we design our framework to invert the generation process. As illustrated in Figure 1, we posit that the observed data $\mathcal{X}$ is a collider generated by two latent confounding factors: the Semantic Factor ($\mathcal{C}$), which governs the clustering-invariant structure, and the Sensitive Factor ($\mathcal{S}$), which encodes bias-specific variations. Traditional anchor-based methods compress $\mathcal{X}$ into a unified anchor set, which inevitably entangles $\mathcal{C}$ and $\mathcal{S}$, creating spurious correlations between the learned representation and sensitive attributes.

To explicitly decouple these factors at the source, we propose a dual-anchor mechanism. Instead of learning a single basis, we introduce two distinct sets of anchors for each view $v$: Semantic anchors $\mathbf{A}_u^{(v)} \in \mathbb{R}^{d_v \times m}$, intended to span the semantic subspace derived from $\mathcal{C}$, and sensitive anchors $\mathbf{A}_s^{(v)} \in \mathbb{R}^{d_v \times m}$, intended to capture the sensitive subspace derived from $\mathcal{S}$.

Consequently, the latent representation is disentangled into a Semantic Graph $\mathbf{Z}_u \in \mathbb{R}^{m \times n}$ and a Sensitive Graph $\mathbf{Z}_s \in \mathbb{R}^{m \times n}$. We model the anchor reconstruction process as follows:

$$\mathcal{L}_{re} = \sum_{v=1}^{V} \left\| \mathbf{X}^{(v)} - (\mathbf{A}_u^{(v)} \mathbf{Z}_u + \mathbf{A}_s^{(v)} \mathbf{Z}_s) \right\|_{\mathrm{F}}^2, \quad (1)$$

which forces the model to explain data variability via two separate channels. To further ensure that the semantic subspace is physically isolated from sensitive bias, we impose a Structural Orthogonality constraint:

$$\mathcal{L}_{orth} = \sum_{v=1}^{V} \left\| \mathbf{A}_u^{(v)^\top} \mathbf{A}_s^{(v)} \right\|_{\mathrm{F}}^2. \quad (2)$$

By minimizing the inner product between semantic and sensitive anchors, we ensure that $\mathbf{A}_u$ captures information strictly orthogonal to the bias captured by $\mathbf{A}_s$. This orthogonality constraint acts as a proxy for blocking the causal path, serving as the foundation for our fairness guarantee.

### 3.3. Statistical Independence via Linearized HSIC

While structural orthogonality provides geometric separation, it does not guarantee that the learned semantic graph $\mathbf{Z}_u$ is statistically free from sensitive information. To ensure rigorous fairness, we employ the Hilbert-Schmidt Independence Criterion (HSIC) to enforce statistical independence between $\mathbf{Z}_u$ and the sensitive attributes $\mathbf{G}$. The standard empirical HSIC is defined as:

$$\text{HSIC}(\mathbf{Z}_u, \mathbf{G}) = \text{Tr}(\mathbf{K}_{\mathbf{Z}_u}\mathbf{H}\mathbf{K}_{\mathbf{G}}\mathbf{H}), \tag{3}$$

where $\mathbf{K}_{\mathbf{Z}_u}$ and $\mathbf{K}_{\mathbf{G}}$ are kernel matrices, and $\mathbf{H} = \mathbf{I} - n^{-1}\mathbf{1}\mathbf{1}^{\top}$. A major bottleneck of standard HSIC is the construction of $n \times n$ kernel matrices, which incurs $\mathcal{O}(n^2)$ complexity, rendering it prohibitive for large-scale datasets.

To address this, we propose a Linearized HSIC strategy tailored for anchor graphs. We adopt a linear kernel for the semantic representation, i.e., $\mathbf{K}_{\mathbf{Z}_u} = \mathbf{Z}_u^{\top}\mathbf{Z}_u$. For the sensitive attributes, we pre-compute the centered sensitive kernel $\tilde{\mathbf{K}}_{\mathbf{G}} = \mathbf{H}\mathbf{G}^{\top}\mathbf{G}\mathbf{H} = \bar{\mathbf{G}}^{\top}\bar{\mathbf{G}}$. By substituting these into the HSIC definition and utilizing the cyclic property of the trace, we derive a primal-form objective:

$$\mathcal{L}_{idep} = \text{Tr}(\mathbf{Z}_u\tilde{\mathbf{K}}_{\mathbf{G}}\mathbf{Z}_u^{\top}). \tag{4}$$

Critically, this linearized formulation avoids the explicit construction of any $n \times n$ matrix. As we will detail in the optimization section, the gradient of this term can be computed with $\mathcal{O}(n)$ complexity, ensuring the scalability of our fairness constraint.

### 3.4. Overall Objective Function

Integrating the generative disentanglement, structural constraints, and statistical independence, the unified loss function of CDAL is as follows:

$$\mathcal{L} = \mathcal{L}_{re} + \alpha\mathcal{L}_{idep} + \beta\mathcal{L}_{orths} + \gamma\mathcal{L}_{sens}, \tag{5}$$

where $\alpha, \beta$ and $\gamma$ are the trade-off hyper-parameters. In addition to the terms derived above, we introduce a sensitive alignment term as follows:

$$\mathcal{L}_{sens} = \left\| \mathbf{G} - \mathbf{W}^{\top}\mathbf{Z}_s \right\|_{\text{F}}^2, \tag{6}$$

which encourages the sensitive graph $\mathbf{Z}_s$ to accurately predict the sensitive attributes $G$ via a linear projection $\mathbf{W} \in \mathbb{R}^{m \times h}$. This supervision ensures that $\mathbf{Z}_s$ indeed focuses on encoding the bias information we intend to strip away from $\mathbf{Z}_u$, thereby facilitating a cleaner disentanglement.

We formulate the overall objective function as follows:

$$\min_{\boldsymbol{\Phi}} \sum_{v=1}^{V} \left\| \mathbf{X}^{(v)} - (\mathbf{A}_u^{(v)}\mathbf{Z}_u + \mathbf{A}_s^{(v)}\mathbf{Z}_s) \right\|_{\text{F}}^2$$
$$+\alpha\,\text{Tr}\left(\mathbf{Z}_u\tilde{\mathbf{K}}_{\mathbf{G}}\mathbf{Z}_u^{\top}\right) + \beta\sum_{v=1}^{V}\left\| \mathbf{A}_u^{(v)\top}\mathbf{A}_s^{(v)} \right\|_{\text{F}}^2$$
$$+\gamma\left\| \mathbf{G} - \mathbf{W}^{\top}\mathbf{Z}_s \right\|_{\text{F}}^2 ,$$
$$\text{s.t. } \mathbf{Z}_u^{\top}\mathbf{1} = \mathbf{1}, \mathbf{Z}_u \geq 0, \mathbf{Z}_s^{\top}\mathbf{1} = \mathbf{1}, \mathbf{Z}_s \geq 0, \mathbf{W}\mathbf{W}^{\top} = \mathbf{I}, \tag{7}$$

where $\boldsymbol{\Phi} = \left\{ \mathbf{A}_u^{(v)}, \mathbf{A}_s^{(v)}, \mathbf{Z}_u, \mathbf{Z}_s, \mathbf{W} \right\}$.

### 3.5. Optimization

To solve the optimization problem in Eq. (7), we propose an efficient alternating iterative algorithm. The objective function is not jointly convex, but it is convex with respect to each variable when others are fixed. Therefore, we update them sequentially.

**Step 1:** Update $\mathbf{W}$. Fixing $\mathbf{Z}_s$ and removing terms unrelated to $\mathbf{W}$, the optimization problem becomes:

$$\min_{\mathbf{W}} \|\mathbf{G} - \mathbf{W}^{\top}\mathbf{Z}_s\|_F^2, \quad \text{s.t. } \mathbf{W}\mathbf{W}^{\top} = \mathbf{I}. \tag{8}$$

This is equivalent to the classic Orthogonal Procrustes problem(Schönemann, 1966):

$$\max_{\mathbf{W}} \text{Tr}(\mathbf{W}^{\top}\mathbf{Z}_s\mathbf{G}^{\top}), \quad \text{s.t. } \mathbf{W}\mathbf{W}^{\top} = \mathbf{I}. \tag{9}$$

Let $\mathbf{U}\boldsymbol{\Sigma}\mathbf{V}^{\top}$ be the SVD of $\mathbf{Z}_s\mathbf{G}^{\top}$. The closed-form solution is given by:

$$\mathbf{W} = \mathbf{U}\mathbf{V}^{\top}. \tag{10}$$

**Step 2:** Update Anchors $\mathbf{A}_u^{(v)}$ and $\mathbf{A}_s^{(v)}$. Taking $\mathbf{A}_u^{(v)}$ as an example, fixing other variables, we minimize the reconstruction and orthogonality terms:

$$\min_{\mathbf{A}_u^{(v)}} \|\mathbf{X}^{(v)} - (\mathbf{A}_u^{(v)}\mathbf{Z}_u + \mathbf{A}_s^{(v)}\mathbf{Z}_s)\|_F^2 + \beta\|\mathbf{A}_u^{(v)\top}\mathbf{A}_s^{(v)}\|_F^2. \tag{11}$$

Setting the derivative w.r.t. $\mathbf{A}_u^{(v)}$ to zero leads to:

$$\mathbf{A}_u^{(v)}(\mathbf{Z}_u\mathbf{Z}_u^{\top}) + \beta\mathbf{A}_u^{(v)}(\mathbf{A}_s^{(v)}\mathbf{A}_s^{(v)\top}) = (\mathbf{X}^{(v)} - \mathbf{A}_s^{(v)}\mathbf{Z}_s)\mathbf{Z}_u^{\top}. \tag{12}$$

This is a standard Sylvester equation in the form $\mathbf{A}\mathbf{X} + \mathbf{X}\mathbf{B} = \mathbf{C}$, which can be solved efficiently to obtain a unique closed-form solution (Bartels & Stewart, 1972). $\mathbf{A}_s^{(v)}$ is updated similarly via a symmetric Sylvester equation.

**Step 3:** Update Sensitive Graph $\mathbf{Z}_s$. Fixing other variables, the optimization for $\mathbf{Z}_s$ can be decomposed into independent sub-problems for each column $\mathbf{z}_i^s$. For the $i$-th sample, we have:

$$\min_{\mathbf{z}_i^s} \mathbf{z}_i^{s\top}\mathbf{H}_s\mathbf{z}_i^s - 2\mathbf{p}_i^{\top}\mathbf{z}_i^s, \quad \text{s.t. } \mathbf{z}_i^{s\top}\mathbf{1} = 1, \mathbf{z}_i^s \geq 0, \tag{13}$$

where $\mathbf{H}_s = \sum_{v=1}^{V} \mathbf{A}_s^{(v)^\top} \mathbf{A}_s^{(v)} + \gamma \mathbf{I}$ and $\mathbf{p}_i = \sum_{v=1}^{V} \mathbf{A}_s^{(v)^\top} (\mathbf{x}_i^{(v)} - \mathbf{A}_u^{(v)} \mathbf{z}_i^u) + \gamma \mathbf{W} \mathbf{g}_i$. This is a standard quadratic programming (QP) problem, which can be solved by using standard solvers.

**Step 4:** Fixing other variables, the optimization w.r.t $\mathbf{Z}_u$ minimizes the reconstruction error regularized by the linearized HSIC term:

$$\min_{\mathbf{Z}_u} \mathcal{J}(\mathbf{Z}_u) = \sum_{v=1}^{V} \|\mathbf{X}^{(v)} - (\mathbf{A}_u^{(v)} \mathbf{Z}_u + \mathbf{A}_s^{(v)} \mathbf{Z}_s)\|_F^2 \\ + \alpha \operatorname{Tr}(\mathbf{Z}_u \tilde{\mathbf{K}}_\mathbf{G} \mathbf{Z}_u^\top), \quad \text{s.t. } \mathbf{Z}_u^\top \mathbf{1} = \mathbf{1}, \mathbf{Z}_u \geq 0, \tag{14}$$

where $\tilde{\mathbf{K}}_\mathbf{G} = \bar{\mathbf{G}}^\top \bar{\mathbf{G}}$. Unlike $\mathbf{Z}_s$, the columns of $\mathbf{Z}_u$ are coupled due to the trace term $\operatorname{Tr}(\mathbf{Z}_u \tilde{\mathbf{K}}_\mathbf{G} \mathbf{Z}_u^\top)$, preventing parallel column-wise optimization. Therefore, we employ Projected Gradient Descent (PGD). The gradient of the objective function is derived as:

$$\nabla \mathcal{J}(\mathbf{Z}_u) = 2 \sum_{v=1}^{V} \mathbf{A}_u^{(v)^\top} \mathbf{A}_u^{(v)} \mathbf{Z}_u + 2\alpha \mathbf{Z}_u \tilde{\mathbf{K}}_\mathbf{G} \\ - 2 \sum_{v=1}^{V} \mathbf{A}_u^{(v)^\top} \left( \mathbf{X}^{(v)} - \mathbf{A}_s^{(v)} \mathbf{Z}_s \right), \tag{15}$$

A naive computation of the gradient term $\mathbf{Z}_u \tilde{\mathbf{K}}_\mathbf{G}$ involves the $n \times n$ matrix $\tilde{\mathbf{K}}_\mathbf{G}$, leading to $\mathcal{O}(n^2)$ complexity. To maintain linear scalability, we exploit the low-rank structure of $\tilde{\mathbf{K}}_\mathbf{G}$ and the associativity of matrix multiplication:

$$\mathbf{Z}_u \tilde{\mathbf{K}}_\mathbf{G} = (\mathbf{Z}_u \bar{\mathbf{G}}^\top) \bar{\mathbf{G}}. \tag{16}$$

We first compute $\mathbf{H} = \mathbf{Z}_u \bar{\mathbf{G}}^\top \in \mathbb{R}^{m \times h}$, and then compute $\mathbf{H} \bar{\mathbf{G}} \in \mathbb{R}^{m \times n}$. Since $m, h \ll n$, this reduces the complexity to $\mathcal{O}(nmh)$.

In each inner iteration $t$, we perform a gradient descent step followed by a projection onto the simplex constraint:

$$\mathbf{Z}_u^{(t+1)} = \mathcal{P}_\mathcal{S} \left( \mathbf{Z}_u^{(t)} - \eta_t \nabla \mathcal{J}(\mathbf{Z}_u^{(t)}) \right), \tag{17}$$

where $\eta_t$ is the step size determined by the Armijo backtracking line search to ensure convergence. $\mathcal{P}_\mathcal{S}(\cdot)$ denotes the Euclidean projection operator that maps each column of the matrix onto the probability simplex. The detailed PGD algorithm, including the specific implementation of the simplex projection and the Armijo line search for step size selection, is provided in Appendix B.

# 4. Theoretical Analysis

In this section, we provide a comprehensive theoretical analysis of CDAL, focusing on its computational complexity, convergence properties, and the theoretical guarantee of causal disentanglement.

## 4.1. Complexity Analysis

The computational cost of our alternating optimization algorithm primarily stems from the updates of four vari-

---

**Algorithm 1** The proposed CDAL Algorithm

**Require:** Multi-view dataset $\{\mathbf{X}^{(v)}\}_{v=1}^{V}$, sensitive attributes $\mathbf{G}$, number of anchors $m$, number of clusters $k$, parameters $\alpha, \beta, \gamma$.
1: Initialize anchors $\mathbf{A}_u^{(v)}, \mathbf{A}_s^{(v)}$ via $k$-means on $\mathbf{X}^{(v)}$, and initialize graphs $\mathbf{Z}_u, \mathbf{Z}_s$ uniformly.
2: **while** not converged **do**
3:     Update $\mathbf{W}$ by solving SVD of $\mathbf{Z}_s \mathbf{G}^\top$.
4:     **for** $v = 1 \to V$ **do**
5:         Update $\mathbf{A}_u^{(v)}$ and $\mathbf{A}_s^{(v)}$ by solving Sylvester Equations (Eq. (12)).
6:     **end for**
7:     Update $\mathbf{Z}_s$ by solving independent QP problems (Eq. (13)).
8:     Update $\mathbf{Z}_u$ via PGD algorithm.
9: **end while**
**Ensure:** Perform $k$-means on the right singular matrix of $\mathbf{Z}_u$ to obtain the final clustering indicators.

---

ables. First, updating the projection matrix $\mathbf{W}$ involves a matrix multiplication with complexity $\mathcal{O}(nmh)$ and an SVD costing $\mathcal{O}(m^2 h)$. Second, for the anchor update, solving the Sylvester equation incurs a complexity of approximately $\mathcal{O}(d^3 + m^3)$, matrix multiplication incurs $\mathcal{O}(nm^2 + m^2 d + nmd)$. Third, the optimization of the sensitive graph $\mathbf{Z}_s$ decouples into $n$ independent quadratic programming sub-problems with a complexity of $\mathcal{O}(nm^3)$, and $\mathcal{O}(nmd + m^2 d)$ for matrix multiplication. Finally, the update of the semantic graph $\mathbf{Z}_u$ costing $\mathcal{O}(nmh + m^2 d + nmd)$ for matrix multiplication and $\mathcal{O}(nm \log m)$ for simplex projection.

Consequently, summing these components, the total time complexity per iteration is dominated by terms linear in $n$, specifically $\mathcal{O}(n(md + mh + m^3))$, where $d = \sum d_v$. This strictly linear complexity contrasts sharply with traditional fair clustering methods that typically scale quadratically ($\mathcal{O}(n^2)$), thereby demonstrating the superior efficiency and scalability of CDAL for large-scale datasets.

## 4.2. Convergence Analysis

The proposed optimization algorithm employs a block coordinate descent strategy. We establish its convergence through the following theorem.

**Theorem 4.1** (Convergence). *The objective function $\mathcal{J}$ in Eq. (7) is lower-bounded by 0. The sequence of objective values $\{\mathcal{J}^{(t)}\}_{t=1}^{\infty}$ generated by our algorithm is monotonically non-increasing, i.e., $\mathcal{J}^{(t+1)} \leq \mathcal{J}^{(t)}$, and converges to a stationary point.*

*Proof.* The objective function consists of non-negative norms and trace terms, ensuring $\mathcal{J} \geq 0$. In each iteration $t$:

1. The sub-problems for $\mathbf{W}$, solved via the Orthogonal Procrustes problem, and for $\mathbf{A}_u^{(v)}, \mathbf{A}_s^{(v)}$, solved via the Sylvester equation, admit closed-form global optima. This ensures $\mathcal{J}(\mathbf{W}^{(t+1)}) \leq \mathcal{J}(\mathbf{W}^{(t)})$, $\mathcal{J}(\mathbf{A}_u^{(v)(t+1)}) \leq \mathcal{J}(\mathbf{A}_u^{(v)(t)})$, and $\mathcal{J}(\mathbf{A}_s^{(v)(t+1)}) \leq \mathcal{J}(\mathbf{A}_s^{(v)(t)})$.

2. The optimization of $\mathbf{Z}_s$ and $\mathbf{Z}_u$ involves convex quadratic programs with simplex constraints. Due to the Lipschitz continuity of the gradient, applying Projected Gradient Descent or QP solvers with a sufficiently small step size ensures a monotonic decrease in the objective function

Therefore, the sequence $\{\mathcal{J}^{(t)}\}$ is monotonic and bounded, implying convergence. □

### 4.3. Identifiability of Disentanglement

A critical theoretical challenge is the identifiability of the learned representations. Standard matrix factorization suffers from rotational ambiguity, where an arbitrary rotation $\mathbf{Q}$ can mix sensitive information back into the semantic representation, rendering fairness ineffective. We prove that CDAL resolves this ambiguity under rigorous causal conditions.

**Theorem 4.2** (Identifiability of Causal Disentanglement). *Let $\mathbf{X} = \mathbf{A}_u^* \mathbf{Z}_u^* + \mathbf{A}_s^* \mathbf{Z}_s^*$ be the true generation process. Assuming statistical independence between factors, non-Gaussianity of $\mathbf{Z}_u^*$, and structural orthogonality between subspaces $(\mathbf{A}_u^{*\top} \mathbf{A}_s^* = \mathbf{0})$, the global minimizer $(\hat{\mathbf{A}}_u, \hat{\mathbf{Z}}_u)$ of CDAL uniquely identifies the true semantic subspace. Specifically, $\hat{\mathbf{Z}}_u = \mathbf{M}_{uu} \mathbf{Z}_u^*$, where $\mathbf{M}_{uu}$ is a non-singular matrix.*

The proof is provided in Appendix A. This theorem implies that the dual-anchor mechanism combined with linearized HSIC physically isolates the causal factors responsible for semantic structure. By ensuring the mixing block between sensitive sources and semantic representations vanishes, CDAL theoretically secures both clustering utility and fairness.

## 5. Experiments

### 5.1. Datasets

We conduct experiments on six fair datasets, including Zafar(Zafar et al., 2017), Har(Anguita et al., 2013), Jaffe(Lyons et al., 1998), COIL(Li et al., 2024), Scene(Li et al., 2024), and Yale(Cai et al., 2007). Har and Jaffe are natural datasets for fair clustering(Li et al., 2024). Specifically, in the Yale dataset, individuals wearing glasses form one protected group, whereas those without glasses form the

*Table 1.* Datasets used in our experiments.

| Dataset | Samples | Features | Protected Groups |
|---------|---------|----------|------------------|
| Yale | 165 | 409/512/50 | w/o glasses (2) |
| Jaffe | 213 | 200/400/600/800/1000/676 | Expression (7) |
| COIL | 1440 | 1024/3304/6750 | Synthetic Binary (2) |
| Scene | 4485 | 20/59/40 | Synthetic Binary (2) |
| Har | 10299 | 400/1000/561 | Person Identity (30) |
| Zafar | 100000 | 200 | Binary (2) |

other protected group. The Zafar dataset is a widely used synthetic dataset, which, similar to the COIL and Scene datasets, generates a binary value as the sensitive attribute. Detailed descriptions of these datasets are provided in Table 1.

### 5.2. Setup

We compare our method with six state-of-the-art fair clustering approaches, including four multi-view fair clustering methods, namely Fair-MVC(Zheng et al., 2023), IMVC(Wang et al., 2025), DFMVC(Zhao et al., 2024), and AKAN(Xu et al., 2025), as well as two single-view fair clustering methods, i.e., SFD(Backurs et al., 2019) and KFC(Harb & Lam, 2020). For single-view fair clustering methods, we concatenate features from all views into a unified representation and perform clustering on this single view. For our method, all hyperparameters are selected from the range $\{10^{-5}, 10^{-3}, 10^{-1}, 10, 10^3\}$, and the number of anchors $m$ is selected from the range $\{k, 2k, 3k, 4k, 5k\}$. The hyperparameters of the competing methods are tuned according to the recommendations provided in their respective papers.

We evaluate clustering performance using Accuracy (ACC), Normalized Mutual Information (NMI), Purity (Pur), and F-score (Fsc). To assess fairness, we adopt two widely used metrics: Balance (Bal) and Minimal Normalized Conditional Entropy (MNCE) (Wei et al., 2025). Details are provided in Appendix C.2. For all metrics, the larger the value is, the better the result is. In addition to reporting individual performance and fairness metrics, we further characterize the trade-off between clustering quality and fairness by plotting the utility-fairness trade-off with respect to ACC and Bal. Since both metrics are maximized, a solution is considered superior utility-fairness trade-off if no other solution achieves higher ACC and Bal simultaneously.

### 5.3. Main Results

Table 2 presents the performance of CDAL compared with six baselines across six benchmark datasets. The results indicate that CDAL effectively balances clustering utility with fairness constraints. On datasets such as Jaffe and Har, CDAL achieves the highest scores in clustering metrics, including ACC, NMI, and Purity, while maintaining

*Table 2.* Clustering and fairness performance comparison on six benchmark datasets. The best and second-best results are highlighted in **bold** and underline. 'TLE': Time Limit Exceeded; 'OS': Out of Scope.

| Method | Yale | | | | | | Jaffe | | | | | |
|---|---|---|---|---|---|---|---|---|---|---|---|---|
| | ACC | NMI | Pur | Fsc | Bal | MNCE | ACC | NMI | Pur | Fsc | Bal | MNCE |
| Fair-MVC | 26.67 | 32.74 | 26.67 | 20.19 | **12.90** | **67.74** | 29.60 | 39.10 | 29.60 | 30.00 | 8.30 | 87.60 |
| SFD | 32.73 | 60.83 | 36.97 | 34.98 | 12.12 | 65.30 | OS | OS | OS | OS | OS | OS |
| KFC | 51.52 | 57.02 | 53.94 | 31.66 | 0.00 | 0.00 | 40.38 | 50.30 | 43.66 | 33.77 | 33.33 | 95.80 |
| DFMVC | 38.18 | 48.55 | 40.61 | 24.85 | 0.00 | 0.00 | 34.27 | 55.85 | 34.74 | 37.71 | 0.00 | 0.00 |
| AKAN | 14.55 | 16.68 | 14.55 | 12.84 | 7.69 | 49.05 | 34.27 | 47.63 | 34.27 | 35.46 | 33.33 | 95.89 |
| IMVC | 55.76 | 64.01 | 58.79 | **43.31** | 0.00 | 0.00 | 64.79 | 77.74 | 69.48 | 63.01 | 0.00 | 35.63 |
| **Ours** | **60.36** | **64.07** | **63.30** | 30.90 | 12.50 | 66.50 | **96.24** | **94.07** | **96.24** | **92.57** | **33.33** | **97.85** |

| Method | COIL | | | | | | Scene | | | | | |
|---|---|---|---|---|---|---|---|---|---|---|---|---|
| | ACC | NMI | Pur | Fsc | Bal | MNCE | ACC | NMI | Pur | Fsc | Bal | MNCE |
| Fair-MVC | 40.35 | 58.96 | 40.35 | 36.85 | 73.68 | 98.34 | 26.85 | 22.67 | 30.43 | 17.31 | 82.00 | 99.30 |
| SFD | 58.54 | 66.62 | 62.15 | 48.54 | **88.99** | **99.76** | 29.99 | 28.03 | 33.89 | 18.89 | 85.02 | 99.53 |
| KFC | 8.82 | 14.46 | 8.82 | 10.99 | 82.91 | 99.37 | 9.16 | 16.00 | 9.16 | 12.92 | **98.54** | **100.00** |
| DFMVC | 34.93 | 45.09 | 37.64 | 30.06 | 63.64 | 96.41 | 30.81 | 34.89 | 37.08 | 23.12 | 83.02 | 99.38 |
| AKAN | 21.46 | 21.43 | 22.29 | 12.90 | 50.00 | 91.83 | 23.95 | 20.43 | 25.75 | 17.70 | 82.65 | 99.35 |
| IMVC | 64.51 | **77.85** | 69.51 | 58.75 | 72.73 | 98.20 | 29.48 | 36.75 | 34.20 | 24.86 | 81.63 | 99.26 |
| **Ours** | **71.17** | 76.43 | **71.27** | **61.02** | 75.00 | 98.53 | **37.65** | **37.66** | **42.08** | **27.06** | 83.04 | 99.38 |

| Method | Har | | | | | | Zafar | | | | | |
|---|---|---|---|---|---|---|---|---|---|---|---|---|
| | ACC | NMI | Pur | Fsc | Bal | MNCE | ACC | NMI | Pur | Fsc | Bal | MNCE |
| Fair-MVC | 35.21 | 29.77 | 37.14 | 40.93 | 12.50 | 97.01 | 98.43 | 88.38 | 98.43 | 96.92 | 68.56 | 99.96 |
| SFD | OS | OS | OS | OS | OS | OS | 50.79 | 2.00 | 50.79 | 53.31 | 68.51 | 99.96 |
| KFC | 27.01 | 13.75 | 27.01 | 29.91 | 11.48 | 96.55 | 50.47 | 28.00 | 50.47 | 66.19 | 59.67 | 97.78 |
| DFMVC | TLE | TLE | TLE | TLE | TLE | TLE | TLE | TLE | TLE | TLE | TLE | TLE |
| AKAN | 36.70 | 49.61 | 36.70 | **47.74** | 0.00 | 24.32 | 94.63 | 72.05 | 94.63 | 89.88 | 65.28 | 99.26 |
| IMVC | 43.36 | 49.04 | 44.06 | 47.15 | 0.00 | 0.00 | **99.93** | **99.14** | **99.93** | **99.85** | 65.34 | 99.28 |
| **Ours** | **75.26** | **68.36** | **75.26** | 66.91 | **14.29** | **98.38** | 98.88 | 91.31 | 98.88 | 97.79 | **68.64** | **99.98** |

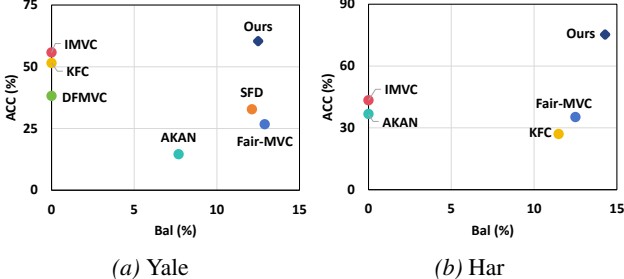

*(a)* Yale  *(b)* Har

*Figure 3.* Trade-off between clustering accuracy (ACC) and balance (Bal) on different datasets.

*Table 3.* Running time comparison (in seconds) on all datasets. 'TLE': Time Limit Exceeded; 'OS': Out of Scope.

| Algorithm | Yale | Jaffe | COIL | Scene | Har | Zafar |
|---|---|---|---|---|---|---|
| Fair-MVC | 17.96 | 17.28 | 110.08 | 102.92 | 362.11 | 543.87 |
| SFD | 8.87 | OS | 74.47 | 14.34 | OS | 70.21 |
| KFC | 0.33 | 1.11 | 43.78 | 3.28 | 27.90 | 7.00 |
| DFMVC | 71.82 | 11.83 | 860.59 | 1073.78 | TLE | TLE |
| AKAN | 27.41 | 121.73 | 238.60 | 419.05 | 979.09 | 3185.21 |
| IMVC | 79.78 | 62.31 | 350.47 | 777.16 | 1395.08 | 2641.91 |
| Ours | 0.75 | 0.57 | 1743.60 | 35.06 | 96.84 | 678.32 |

competitive performance in fairness metrics like Balance and MNCE. A notable comparison is observed on the Zafar dataset. While the baseline SFD suffers a significant

decline in NMI to 2.00% to satisfy fairness requirements, CDAL maintains a high NMI of 91.31% alongside a Balance score of 68.64%. These results suggest that the proposed structural disentanglement mechanism can filter out sensitive information without discarding the semantic structures essential for clustering.

*Table 4.* Ablation study on six datasets. We incrementally add the orthogonality constraint ($\mathcal{L}_{orth}$), independence constraint ($\mathcal{L}_{indep}$), and sensitive alignment ($\mathcal{L}_{sens}$) to evaluate their contributions. The best combination (Ours) is highlighted in gray.

| Components | | | Yale | | Jaffe | | COIL | | Scene | | Har | | Zafar | |
|---|---|---|---|---|---|---|---|---|---|---|---|---|---|---|
| $\mathcal{L}_{orth}$ | $\mathcal{L}_{indep}$ | $\mathcal{L}_{sens}$ | ACC | Bal | ACC | Bal | ACC | Bal | ACC | Bal | ACC | Bal | ACC | Bal |
| – | – | – | 74.27 | 0.00 | 51.57 | 0.00 | 62.56 | 54.55 | 20.55 | 86.49 | 38.45 | 0.00 | 51.20 | 68.45 |
| – | – | ✓ | 74.27 | 0.00 | 54.30 | 0.00 | 62.56 | 54.55 | 20.48 | 86.84 | 38.48 | 0.00 | 50.77 | 67.81 |
| – | ✓ | – | 79.06 | 0.00 | 50.85 | 0.00 | 62.09 | 38.46 | 20.56 | 86.49 | 38.44 | 0.00 | 51.20 | 68.45 |
| ✓ | – | – | 60.82 | 8.33 | 60.75 | 0.00 | 62.56 | 54.55 | 20.55 | 86.49 | 44.96 | 0.00 | 74.98 | 65.68 |
| – | ✓ | ✓ | 79.06 | 0.00 | 55.59 | 0.00 | 62.09 | 38.46 | 20.48 | 86.84 | 38.49 | 0.00 | 50.77 | 67.81 |
| ✓ | – | ✓ | 60.82 | 8.33 | 61.64 | 0.00 | 62.56 | 54.55 | 20.48 | 86.84 | 44.95 | 0.00 | 74.98 | 65.68 |
| ✓ | ✓ | – | 60.82 | 8.33 | 67.91 | 0.00 | 62.09 | 38.46 | 20.55 | 86.49 | 45.00 | 0.00 | 74.98 | 65.68 |
| ✓ | ✓ | ✓ | 60.36 | 12.50 | 96.24 | 33.33 | 71.17 | 75.00 | 37.65 | 83.04 | 75.26 | 14.29 | 98.88 | 68.64 |

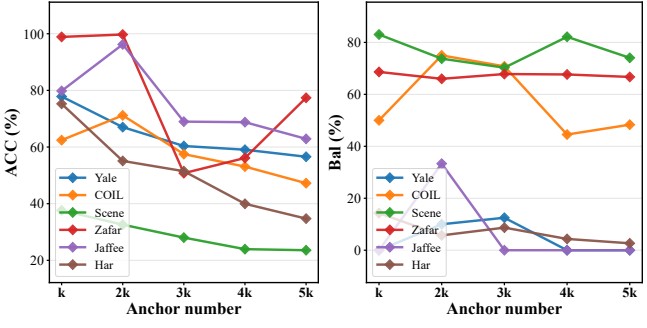

*(a)* ACC vs. Anchor Number    *(b)* Balance vs. Anchor Number

*(a)* Yale    *(b)* Har

*Figure 4.* Parameter sensitivity analysis w.r.t. the number of anchors $m$.

*Figure 5.* Objective value curves w.r.t the number of iterations.

To visually evaluate the relationship between clustering quality and fairness, we plot the utility-fairness trade-off in Fig. 3. Here, the y-axis represents clustering accuracy and the x-axis denotes fairness measured by Balance. Ideally, a method should be located in the top-right region of the plot. As illustrated on representative datasets such as Yale and Har, baseline methods typically exhibit a clear compromise, appearing either in the top-left region characterized by high utility and low fairness or the bottom-right region with low utility and high fairness. In contrast, CDAL, denoted by a red star, consistently locates in the upper-right quadrant. This positioning indicates that our dual-anchor approach effectively mitigates the inherent conflict between the two objectives, achieving improved fairness with minimal loss in clustering accuracy compared to existing approaches.

### 5.4. Running Time Comparison

Table 3 reports the running time comparison of different methods across all datasets. The results highlight the superior scalability of the proposed framework. While heuristic baselines like KFC and SFD exhibit lower runtimes on some

datasets due to their algorithmic simplicity, they often suffer from severe performance degradation, as evidenced in Table 2. Crucially, when compared to state-of-the-art fair multi-view methods (e.g., DFMVC, AKAN, and IMVC), CDAL demonstrates a significant efficiency advantage. On large-scale benchmarks such as Har and Zafar, these complex baselines frequently encounter Time Limit Exceeded (TLE) failures or incur prohibitive computational costs. In contrast, CDAL consistently completes optimization within a reasonable time budget, benefiting from its linear time complexity $\mathcal{O}(n)$. This confirms that CDAL is the most practical solution for large-scale applications, striking an optimal balance between computational efficiency and clustering performance.

### 5.5. Ablation Study

To validate the contribution of each component in our proposed framework, we conducted a comprehensive ablation study by incrementally incorporating the structural orthogonality constraint ($\mathcal{L}_{orth}$), the linearized HSIC independence constraint ($\mathcal{L}_{indep}$), and the sensitive anchor alignment loss ($\mathcal{L}_{sens}$), as reported in Table 4. We observe that the baseline

method consistently fails to ensure fairness, often yielding a Balance score of zero on datasets like Yale and Har, which highlights the inherent bias in unconstrained multi-view clustering. While applying $\mathcal{L}_{orth}$ or $\mathcal{L}_{indep}$ in isolation offers limited gains, the complete CDAL framework demonstrates significant superiority by achieving the highest accuracy and fairness scores across all benchmarks.

### 5.6. Sensitivity Analysis

We investigate the impact of the anchor number $m$ on clustering performance in Fig. 4. The results indicate that performance generally peaks when $m$ is small. Increasing $m$ further tends to degrade accuracy due to the introduction of noise and redundancy, while fairness metrics remain relatively stable. Additional sensitivity analyses for regularization hyperparameters ($\alpha, \beta, \gamma$) are provided in the Appendix due to space limitations.

### 5.7. Convergence Analysis

To empirically validate the convergence properties of our optimization algorithm, we track the objective function values on the Yale and Har datasets. As illustrated in Fig. 5, the objective value exhibits a rapid and monotonic decrease, stabilizing within 50 iterations on both datasets. This observation is consistent with our theoretical analysis, confirming the efficiency and numerical stability of the proposed alternating optimization scheme in practice.

## 6. Conclusion

In this paper, we addressed the challenging problem of fairness in multi-view clustering by proposing a novel framework, Causal Disentangled Anchor Learning (CDAL). Unlike existing methods that rely on implicit correlation suppression, CDAL introduces a dual-anchor mechanism to explicitly separate latent representations into orthogonal semantic and sensitive subspaces from a structural causal perspective. Theoretical analysis guarantees the identifiability of the disentangled factors and the linear time complexity of our algorithm. Extensive experiments on benchmarks demonstrate that CDAL significantly mitigates the conflict between clustering utility and fairness, achieving a superior trade-off compared to state-of-the-art baselines.

## Acknowledgment

This work is supported by the National Science Fund for Distinguished Young Scholars of China (No. 62325604), the National Natural Science Foundation of China (No. 625B2182, 62441618 and 62276271), the Major Program Project of Xiangjiang Laboratory (No. 24XJJCYJ01002), and the Provincial Natural Science Foundation of Hunan (No. 2025JJ10008).

## Impact Statement

This paper presents a method for fair multi-view clustering. Our work aims to mitigate bias in machine learning models, contributing to more equitable algorithmic decision-making. We do not foresee any immediate negative societal consequences, although like all ML methods, it should be deployed with care in sensitive domains.

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

# A. Proof of Identifiability of Disentanglement

In this section, we provide the detailed proof for Theorem 4.2. We aim to show that under specific structural and statistical assumptions, the proposed CDAL framework achieves identifiability of disentanglement. Specifically, we prove that the framework effectively eliminates the rotation ambiguity inherent in matrix factorization, ensuring that the learned semantic representations are completely disentangled from sensitive factors.

## A.1. Problem Setup and Assumptions

Let the true data generation process be described by the structural causal model:

$$\mathbf{X} = \mathbf{A}_u^* \mathbf{Z}_u^* + \mathbf{A}_s^* \mathbf{Z}_s^*, \tag{18}$$

where $\mathbf{A}_u^*, \mathbf{A}_s^*$ are the true ground-truth anchor matrices, and $\mathbf{Z}_u^*, \mathbf{Z}_s^*$ are the true latent semantic and sensitive factors, respectively. The model learns estimates $\hat{\mathbf{A}}_u, \hat{\mathbf{A}}_s, \hat{\mathbf{Z}}_u, \hat{\mathbf{Z}}_s$.

We make the following standard assumptions for identifiability:

1. The true semantic graph $\mathbf{Z}_u^*$ and sensitive graph $\mathbf{Z}_s^*$ are statistically independent.

2. The components of $\mathbf{Z}_u^*$ follow a non-Gaussian distribution.

3. The true subspaces are orthogonal, i.e., $\mathbf{A}_u^{*\top} \mathbf{A}_s^* = \mathbf{0}$, and basis matrices have full column rank.

## A.2. Proof of Theorem 4.2

**Theorem 4.2.** *Under the above assumptions, the global minimizer $(\hat{\mathbf{Z}}_u, \hat{\mathbf{A}}_u)$ of the CDAL objective function uniquely identifies the true semantic subspace. That is, the learned semantic representation is a linear transformation of the true semantic factors, strictly isolated from the sensitive factors: $\hat{\mathbf{Z}}_u = \mathbf{M}_{uu} \mathbf{Z}_u^*$, where $\mathbf{M}_{uu}$ is a non-singular matrix.*

*Proof.* Matrix factorization suffers from rotational ambiguity. For any invertible matrix $\mathbf{Q}$, we can write:

$$\mathbf{X} = [\mathbf{A}_u^*, \mathbf{A}_s^*] \begin{bmatrix} \mathbf{Z}_u^* \\ \mathbf{Z}_s^* \end{bmatrix} = [\mathbf{A}_u^*, \mathbf{A}_s^*] \mathbf{Q} \mathbf{Q}^{-1} \begin{bmatrix} \mathbf{Z}_u^* \\ \mathbf{Z}_s^* \end{bmatrix}. \tag{19}$$

Our goal is to prove that the constraints in CDAL force the inverse mixing matrix $\mathbf{Q}^{-1}$ to assume a block-structure that prevents the mixing of $\mathbf{Z}_s^*$ into $\hat{\mathbf{Z}}_u$.

Let the inverse mixing matrix be $\mathbf{M} = \mathbf{Q}^{-1}$, partitioned as:

$$\begin{bmatrix} \hat{\mathbf{Z}}_u \\ \hat{\mathbf{Z}}_s \end{bmatrix} = \begin{bmatrix} \mathbf{M}_{uu} & \mathbf{M}_{us} \\ \mathbf{M}_{su} & \mathbf{M}_{ss} \end{bmatrix} \begin{bmatrix} \mathbf{Z}_u^* \\ \mathbf{Z}_s^* \end{bmatrix}. \tag{20}$$

First, the objective minimizes $\mathcal{L}_{indep} = \alpha \text{HSIC}(\hat{\mathbf{Z}}_u, \mathbf{G})$, enforcing independence between $\hat{\mathbf{Z}}_u$ and the sensitive source $\mathbf{Z}_s^*$. From the expansion $\hat{\mathbf{Z}}_u = \mathbf{M}_{uu} \mathbf{Z}_u^* + \mathbf{M}_{us} \mathbf{Z}_s^*$, we observe that $\hat{\mathbf{Z}}_u$ is a linear combination of independent non-Gaussian sources (Assumption 1 & 2). Since $\hat{\mathbf{Z}}_u$ is forced to be independent of $\mathbf{Z}_s^*$, the mixing coefficient corresponding to $\mathbf{Z}_s^*$ must vanish. Thus, $\mathbf{M}_{us} = \mathbf{0}$, we have $\hat{\mathbf{Z}}_u = \mathbf{M}_{uu} \mathbf{Z}_u^*$. This confirms that $\hat{\mathbf{Z}}_u$ contains no information from the sensitive subspace.

To validate the representation, we must ensure $\mathbf{M}_{uu}$ is non-singular. The CDAL objective imposes a structural orthogonality constraint $\|\hat{\mathbf{A}}_u^\top \hat{\mathbf{A}}_s\|_F^2$ and minimizes reconstruction error $\|\mathbf{X} - \hat{\mathbf{A}}\hat{\mathbf{Z}}\|_F^2$. Given that the true generators are orthogonal (Assumption 3), any rank deficiency in $\mathbf{M}_{uu}$ would imply a loss of semantic information that cannot be compensated by the sensitive branch. Therefore, to minimize reconstruction error, the optimizer must maintain a full-rank transformation $\mathbf{M}_{uu}$ to span the semantic subspace defined by $\mathbf{Z}_u^*$.

Therefore, we conclude that $\hat{\mathbf{Z}}_u = \mathbf{M}_{uu} \mathbf{Z}_u^*$ with a non-singular $\mathbf{M}_{uu}$. The framework successfully recovers the true semantic subspace while guaranteeing fairness via disentanglement. $\square$

---

**Algorithm 2** Optimization for Semantic Graph $\mathbf{Z}_u$

---

**Require:** Current $\mathbf{Z}_u$, Anchors $\mathbf{A}_u, \mathbf{A}_s$, Residuals $\mathbf{R}^{(v)}$, Sensitive Data $\bar{\mathbf{G}}$.

 1: **while** not converged **do**
 2:    Compute $\nabla \mathcal{J}(\mathbf{Z}_u)$ with Eq (22).
 3:    **Line Search:** Initialize $\eta = 1.0$.
 4:    **repeat**
 5:       Compute candidate $\mathbf{Z}_{temp} = \mathbf{Z}_u - \nabla \mathcal{J}(\mathbf{Z}_u)$.
 6:       Project onto simplex $\mathbf{Z}_{new} = \mathcal{P}_{\mathcal{S}}(\mathbf{Z}_{temp})$ via sorting algorithm.
 7:       **if** Armijo condition (Eq. (23)) is not met **then**
 8:          Update $\eta \leftarrow 0.5\eta$.
 9:       **end if**
10:    **until** Armijo condition is satisfied
11:    Update $\mathbf{Z}_u \leftarrow \mathbf{Z}_{new}$.
12: **end while**

---

## B. Optimization Details for Semantic Graph $\mathbf{Z}_u$

In this section, we provide the detailed derivation and algorithm for updating the semantic graph $\mathbf{Z}_u$. This sub-problem involves a non-convex objective with simplex constraints, which we solve using a Projected Gradient Descent (PGD) scheme characterized by a linear-complexity gradient computation and an Armijo backtracking line search.

Fixing anchors $\{\mathbf{A}_u^{(v)}, \mathbf{A}_s^{(v)}\}_{v=1}^V$ and the sensitive graph $\mathbf{Z}_s$, the optimization problem with respect to $\mathbf{Z}_u$ is formulated as follows:

$$\min_{\mathbf{Z}_u} \mathcal{J}(\mathbf{Z}_u) = \sum_{v=1}^V \|\mathbf{X}^{(v)} - (\mathbf{A}_u^{(v)}\mathbf{Z}_u + \mathbf{A}_s^{(v)}\mathbf{Z}_s)\|_F^2 + \alpha \operatorname{Tr}(\mathbf{Z}_u \mathbf{K_G} \mathbf{Z}_u^\top),$$
$$\text{s.t. } \mathbf{Z}_u^\top \mathbf{1} = \mathbf{1}, \mathbf{Z}_u \geq 0, \tag{21}$$

### B.1. Gradient Computation with Linear Complexity

The gradient of the reconstruction term is straightforward. The primary computational challenge lies in the linearized HSIC term $\operatorname{Tr}(\mathbf{Z}_u \mathbf{K_G} \mathbf{Z}_u^\top)$, where $\mathbf{K_G} = \bar{\mathbf{G}}^\top \bar{\mathbf{G}} \in \mathbb{R}^{n \times n}$ is the centered sensitive kernel. A direct computation of $\mathbf{Z}_u \mathbf{K_G}$ would incur $\mathcal{O}(n^2 m)$ complexity, undermining the scalability of our framework.

To ensure the $\mathcal{O}(n)$ scalability of CDAL, we leverage the low-rank factorization of $\mathbf{K_G}$. The full gradient is computed as:

$$\nabla \mathcal{J}(\mathbf{Z}_u) = 2 \sum_{v=1}^V \mathbf{A}_u^{(v)^\top} \mathbf{A}_u^{(v)} \mathbf{Z}_u - 2 \sum_{v=1}^V \mathbf{A}_u^{(v)^\top}(\mathbf{X}^{(v)} - \mathbf{A}_s^{(v)}\mathbf{Z}_s) + 2\alpha \mathbf{Z}_u \bar{\mathbf{G}}^\top \bar{\mathbf{G}}. \tag{22}$$

We compute $\mathbf{H} = \mathbf{Z}_u \bar{\mathbf{G}}^\top$ with $\mathcal{O}(nmh)$ cost at first, and compute the gradient component $\mathbf{H}\bar{\mathbf{G}}$ with $\mathcal{O}(nmh)$ cost then. This strategy avoids constructing any $n \times n$ matrix, keeping the memory and time consumption linear with respect to $n$.

### B.2. Projected Gradient Descent Algorithm

We employ an iterative update rule $\mathbf{Z}_u^{(t+1)} = \mathcal{P}_{\mathcal{S}}(\mathbf{Z}_u^{(t)} - \eta_t \nabla \mathcal{J}(\mathbf{Z}_u^{(t)}))$ to solve the constrained optimization problem. The procedure integrates an efficient simplex projection operator and an adaptive line search strategy to ensure convergence.

**Efficient Simplex Projection.** The operator $\mathcal{P}_{\mathcal{S}}(\mathbf{V})$ projects each column of a matrix $\mathbf{V}$ onto the probability simplex. Mathematically, for a vector $\mathbf{v}$, it solves $\min_{\mathbf{p}} \|\mathbf{p} - \mathbf{v}\|_2^2$ subject to $\mathbf{p} \geq 0, \mathbf{1}^\top \mathbf{p} = 1$. Instead of solving a quadratic program, we utilize the efficient sorting-based algorithm proposed by Duchi et al. (Duchi et al., 2008). This method determines the exact projection by sorting the vector elements to find an optimal threshold, achieving a complexity of $\mathcal{O}(m \log m)$.

**Armijo Backtracking Line Search.** To guarantee a sufficient decrease of the objective function without manual tuning of the learning rate, we adopt the Armijo backtracking rule. At each iteration, starting from an initial step size $\eta$, we iteratively shrink $\eta$ by a factor $\beta \in (0, 1)$ until the following condition is met:

$$\mathcal{J}(\mathbf{Z}_u^{(t+1)}) \leq \mathcal{J}(\mathbf{Z}_u^{(t)}) + \sigma \langle \nabla \mathcal{J}(\mathbf{Z}_u^{(t)}), \mathbf{Z}_u^{(t+1)} - \mathbf{Z}_u^{(t)} \rangle, \tag{23}$$

where $\sigma$ is a small control parameter. The complete optimization procedure for $\mathbf{Z}_u$ is summarized in Algorithm 2.

# C. Supplementary Experiments

## C.1. Datasets

- **Zafar**(Zafar et al., 2017):The Zafar dataset is a widely adopted synthetic benchmark, wherein a single binary attribute is synthesized to serve as the sensitive feature.

- **Har**(Anguita et al., 2013): Har is a single-view facial image dataset; three views (the original view and two auto-encoded views) are constructed, and protected groups are assigned by sampling a Bernoulli distribution with $p = 0.5$.

- **Jaffe**(Lyons et al., 1998): Jaffe is a single-view Japanese facial expression dataset; five additional views are generated using pre-trained ResNet-50, VGG-16, and autoencoders with various hidden-layer sizes, yielding a total of six views for fairness experiments.

- **COIL**[1]: COIL is a multi-view object image dataset; each instance is randomly labeled as belonging to one of two protected groups according to a Bernoulli$(0.5)$ distribution.

- **Scene**[2]: Scene is a multi-view scene-classification dataset; protected groups are created by randomly assigning instances with probability $0.5$ (Bernoulli distribution).

- **Yale**(Cai et al., 2007): Yale is a multi-view face image dataset; protected groups are defined by the presence (glasses) or absence (no glasses) of eyewear, following the convention in (Li et al., 2020).

## C.2. Fairness Metrics

Following prior work(Wei et al., 2025), we adopt Balance (Bal) and Minimal Normalized Conditional Entropy (MNCE) to assess fairness in clustering.

**Balance (Bal).** The Balance metric measures the proportional representation of protected groups within clusters. For a cluster $c$, let $n_c^{\min}$ and $n_c^{\max}$ denote the minimum and maximum numbers of instances among all protected groups in $c$, respectively. The balance score is defined as

$$\text{Bal} = \min_c \frac{n_c^{\min}}{n_c^{\max}} \in [0, 1]. \tag{24}$$

**Minimal Normalized Conditional Entropy (MNCE).** MNCE evaluates fairness by measuring the uniformity of protected group distributions within clusters from an information-theoretic perspective. Let $\mathcal{G}$ denote the set of protected groups. For a cluster $c$, let $p_c(g)$ denote the proportion of samples belonging to group $g \in \mathcal{G}$ in cluster $c$, and let $p(g)$ denote the overall proportion of group $g$ in the entire dataset. MNCE is defined as

$$\text{MNCE} = \frac{\min_c \left( -\sum_{g \in \mathcal{G}} p_c(g) \log p_c(g) \right)}{-\sum_{g \in \mathcal{G}} p(g) \log p(g)} \in [0, 1]. \tag{25}$$

**Remark.** For both Bal and MNCE, larger values indicate better fairness.

## C.3. Details of Compared Methods

- **Fair-MVC**(Zheng et al., 2023): A fairness-aware multi-view clustering method that enforces group balance by incorporating fairness constraints into soft cluster assignments.

- **IMVC**(Wang et al., 2025): A fair incomplete multi-view clustering approach that aligns feature distributions across views to mitigate bias caused by missing data.

---

[1] https://www.cs.columbia.edu/CAVE/software/softlib/coil-20.php
[2] https://figshare.com/articles/dataset/15-Scene_Image_Dataset/7007177

- **DFMVC**(Zhao et al., 2024): A deep fair multi-view clustering method that leverages contrastive learning and distribution alignment to balance fairness and clustering performance.

- **AKAN**(Xu et al., 2025): A deep fair multi-view clustering framework that employs attention-based Kolmogorov–Arnold networks to capture nonlinear inter-view relationships.

- **SFD**(Backurs et al., 2019): A scalable fair clustering algorithm based on efficient fairlet decomposition to ensure balanced cluster compositions.

- **KFC**(Harb & Lam, 2020): A k-center fair clustering algorithm that provides approximation guarantees while preventing over- or under-representation of protected groups.

## C.4. Trade-off on More Datasets

To provide a comprehensive evaluation, we extend the utility-fairness trade-off analysis to the remaining four datasets, including Jaffe, COIL, Scene, and Zafar, as visualized in Fig. 6. Consistent with the main results, CDAL consistently occupies the optimal top-right region across all benchmarks to demonstrate a superior Pareto frontier. Specifically, baseline methods often suffer from catastrophic utility degradation to satisfy fairness constraints, exemplified by KFC on the Zafar dataset where accuracy drops below 50%. Conversely, other approaches fail to ensure fairness despite high accuracy, such as IMVC on Jaffe which yields near-zero Balance scores. In contrast, CDAL successfully mitigates this conflict by maintaining state-of-the-art clustering accuracy while achieving competitive or superior fairness scores. These results further validate the robustness of our structural causal disentanglement mechanism across diverse data distributions.

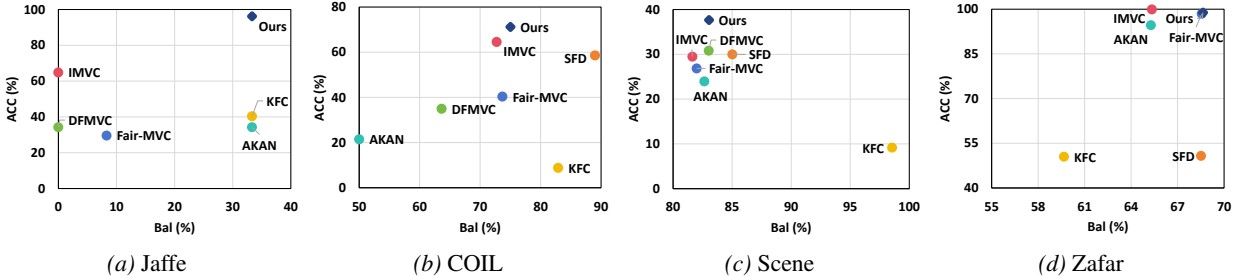

|            |           |           |           |
|:----------:|:---------:|:---------:|:---------:|
| *(a)* Jaffe | *(b)* COIL | *(c)* Scene | *(d)* Zafar |

*Figure 6.* Trade-off between clustering accuracy (ACC) and balance (Bal) on other datasets.

## C.5. Sensitivity Study

We study the sensitivity of the proposed method to the regularization hyperparameters $\alpha$, $\beta$, and $\gamma$. Fig. 7 reports the clustering accuracy (ACC) and balance (Bal) by varying one parameter at a time while fixing the others.

**Effect of $\alpha$.** The method exhibits stable performance over a wide range of $\alpha$ values. ACC remains high for small to moderate settings, while excessively large $\alpha$ may slightly degrade performance on certain datasets. The fairness metric Bal shows limited variation, indicating low sensitivity to $\alpha$.

**Effect of $\beta$.** The parameter $\beta$ controls the strength of the fairness-aware regularization. Moderate values generally achieve a favorable trade-off between ACC and Bal. When $\beta$ becomes overly large, over-regularization may reduce ACC, whereas Bal remains relatively stable across most settings.

**Effect of $\gamma$.** The parameter $\gamma$ governs the causal disentanglement regularization. ACC is largely insensitive to $\gamma$ within a reasonable range, with noticeable degradation only at extreme values. Bal follows a smooth trend overall, demonstrating robustness to the choice of $\gamma$.

Overall, these results indicate that the proposed method is robust to hyperparameter variations and can achieve stable performance without extensive tuning.

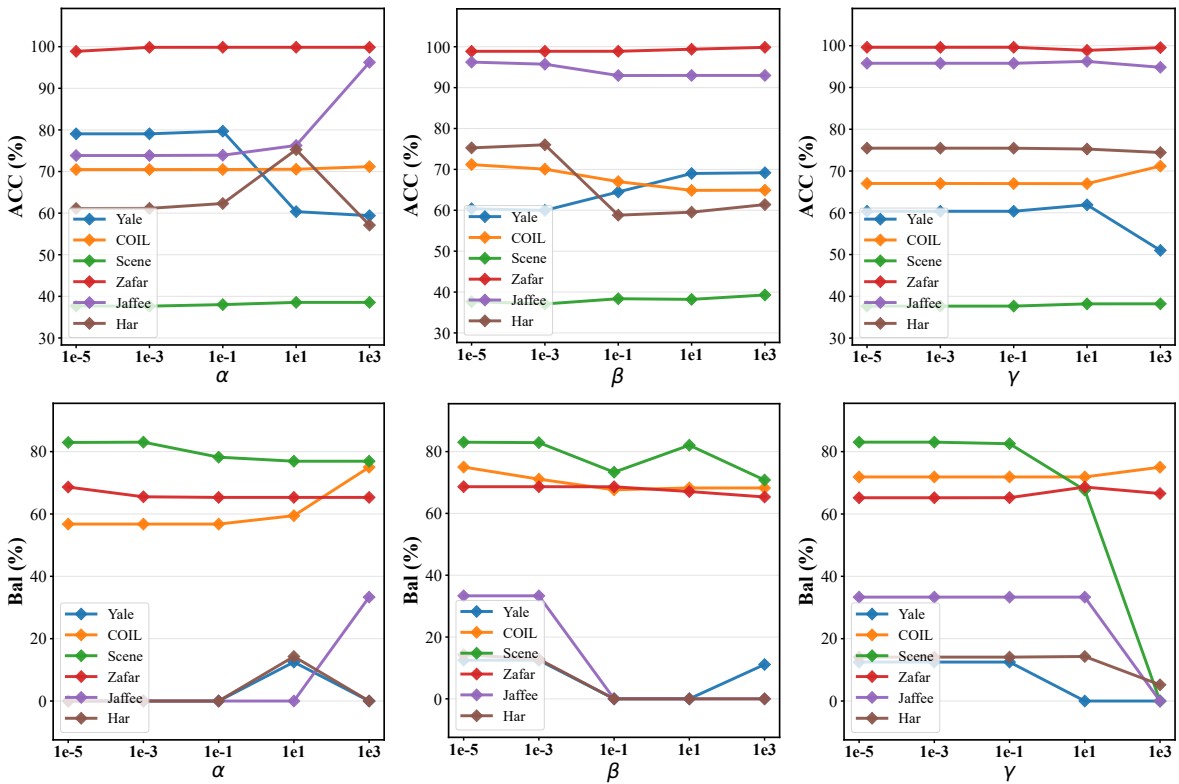

*Figure 7.* Sensitivity analysis w.r.t. parameters $\alpha$, $\beta$, and $\gamma$ in terms of clustering accuracy (ACC) and balance (Bal).

## C.6. Convergence Analysis

To further validate the numerical stability and efficiency of our optimization algorithm, we provide the convergence curves for the objective function on the remaining four datasets. As illustrated in Fig. 8, the objective function values exhibit a sharp and monotonic decline during the initial optimization phase across all datasets. Specifically, the curves stabilize rapidly, typically reaching a steady state within the first 20 to 30 iterations. This consistent convergence behavior empirically confirms that our alternating optimization scheme efficiently converges to a local optimum, ensuring practical scalability even on complex multi-view datasets.

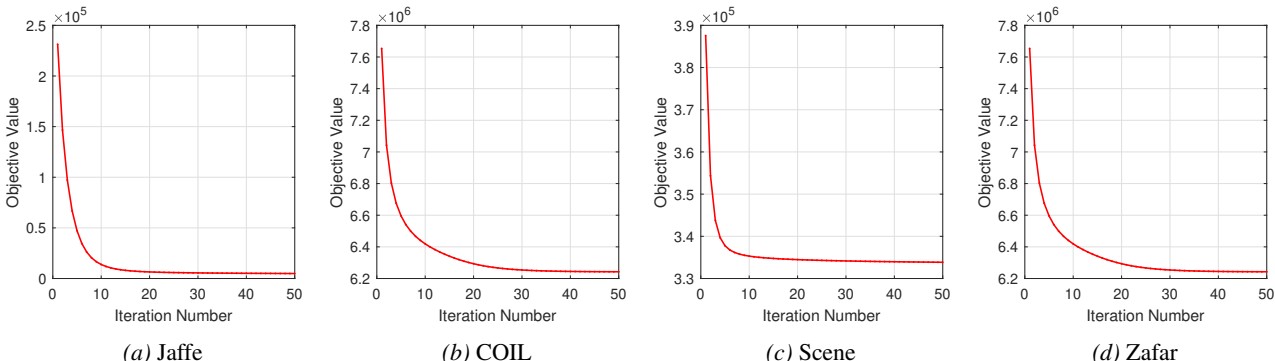

*Figure 8.* Objective value curves w.r.t the number of iterations on other datasets.

