# OpenReview forum: "Causal Disentangled Anchor Learning for Scalable Fair Multi-view Clustering"
_ICML.cc/2026/Conference — ICML 2026 regular_

### Official Review · Reviewer_RBB9 · 2026-03-04

**Soundness:** 3
**Presentation:** 3
**Significance:** 4
**Originality:** 4
**Overall Recommendation:** 4
**Confidence:** 4

**Summary:**

The paper proposes a new framework named CDAL for scalable and fair multi-view clustering. Guided by a Structural Causal Model, CDAL uses a dual-anchor mechanism to structurally separate latent representations into orthogonal semantic and sensitive subspaces. It ensures statistical independence via a linearized HSIC constraint, reducing time complexity to $\mathcal{O}(n)$. Theoretical proofs for identifiability and extensive experiments demonstrate a superior trade-off between clustering utility and fairness.

**Compliance With Llm Reviewing Policy:**

Affirmed.

**Final Justification:**

The authors have address my concern and I would keep my rating.

**Key Questions For Authors:**

1. In Eq. (1), reconstructing samples using semantic and sensitive anchors ($A_u Z_u + A_s Z_s$) introduces approximation errors. Why not directly learn the disentangled latent representations first, and then learn the anchors and the anchor graph from them?

2. The dual-anchor mechanism is an important innovation of this paper. If the framework did not use the structural orthogonality constraint ($\mathcal{L}_{orth}$) to separate the subspaces, and instead relied purely on HSIC independence applied directly to the anchors, would it cause a performance degradation? Please conduct an experiment or provide analysis to clarify this.

3. Algorithm 1 notes that k-means is applied to $Z_u$ to obtain the final clustering labels. Given that $Z_u$ is a non-negative semantic representation representing probabilities, would a direct index extraction (e.g., argmax) or spectral discretization strategy be more robust than k-means?

4. In the paper's time comparison experiments (Table 4) , the running time comparison with classic and recent methods (like SFD and IMVC) clearly demonstrates the high efficiency of the proposed Algorithm. Could the authors briefly discuss if this time efficiency comes at any hidden cost regarding peak memory usage during the alternating optimization steps?

**Limitations:**

While the Impact Statement briefly mentions deploying with care in sensitive domains, the technical limitations of the approach are not fully discussed. The paper would benefit from a brief discussion on scenarios where the method might fall short, such as when the non-Gaussianity assumption for identifiability is violated, or how the approximation error from anchor-based reconstruction scales with the number of views.

**Strengths And Weaknesses:**

Strengths:
1. The linearized HSIC derivation reduces complexity from $\mathcal{O}(n^2)$ to $\mathcal{O}(n)$, making the method highly scalable for large datasets.
2. The theoretical proofs for identifiability under structural orthogonality provide a solid mathematical foundation for the framework's disentanglement claims.

Weakness:
1. Reconstructing samples using anchors ($X \approx A_u Z_u + A_s Z_s$) inherently introduces approximation errors, which may impact the purity of the learned disentanglement on highly complex manifolds.
2. The empirical evaluation lacks statistical transparency, as it fails to specify the number of independent runs and whether the reported results in the main tables denote maximum or average values.

---

> ### Author Rebuttal · Authors · 2026-03-29
>
> **W1:** We acknowledge that dual-anchor reconstruction introduces approximation errors. However, for large-scale multi-view clustering, this is a highly advantageous trade-off. By projecting data onto a low-dimensional subspace ($m \ll n$), anchors can filter out instance-level noise and complex spurious correlations. Crucially, it allows us to enforce causal disentanglement at the source rather than the sample level, preventing sensitive bias from entangling with the semantic structure during large-scale graph construction. Empirical results on large-scale datasets confirm the purity gained from this structural isolation far outweighs the approximation error.
>
> **W2:** All reported results in the main tables denote the average values. Specifically, because the final clustering step relies on standard $k$-means (which is sensitive to initial centroids), we run the $k$-means algorithm 20 independent times on the learned semantic representation $Z_u$ and report the average performance.
>
> **Q1:** We do not directly learn the disentangled latent representation $H \in \mathbb{R}^{d \times n}$ for all $n$ samples first due to computational bottlenecks. Enforcing statistical independence (via HSIC) or structural fairness directly on $H$ requires computing $n \times n$ kernel matrices, driving complexity to $\mathcal{O}(n^2)$. By learning dual anchors ($A_u, A_s$) first, we execute disentanglement in an $\mathcal{O}(m)$ subspace. The anchor graph $Z$ merely acts as a bipartite transition matrix broadcasting this purified representation to $n$ samples, maintaining the required $\mathcal{O}(n)$ complexity.
>
> **Q2:** Relying purely on HSIC without structural orthogonality ($\mathcal{L}_{orth}$) significantly degrades performance. Theoretically, HSIC enforces statistical independence, whereas structural orthogonality geometric separation. Without constraining anchors, bases might be statistically independent but still span overlapping linear subspaces.  During reconstruction, this geometric overlap allows bias to leak into the semantic graph $Z_u$ and structural orthogonality prevents this. This is empirically validated in our ablation study (Table 3): setting $\alpha=0$ causes a noticeable drop in the Balance score, confirming that statistical independence cannot replace structural isolation.
>
> **Q3:** As shown in **Table E**, standard $k$-means for post-processing is essential.
>
> **Table E: $k$-means vs. Direct Argmax**
> | Post-processing Strategy | Metric | Yale | Jaffe | COIL | Scene | Har | Zafar |
> | --- | --- | --- | --- | --- | --- | --- | --- |
> | K-means (Ours) | ACC (%) | 60.36 | 96.24 | 71.17 | 37.65 | 75.26 | 98.88 |
> |  | Bal (%) | 12.50 | 33.33 | 75.00 | 83.04 | 14.29 | 68.64 |
> | Direct Argmax | ACC (%) | 59.39 | 47.42 | 53.26 | 15.23 | 41.97 | 37.70 |
> |  | Bal (%) | 0.00 | 0.00 | 65.96 | 70.94 | 0.00 | 64.91 |
>
> Directly extracting labels via argmax causes severe accuracy degradation (e.g., 98.88% to 37.70% on Zafar) and collapses fairness on multiple datasets (Balance drops to 0.00% on Yale, Jaffe, Har). The raw graph $Z_u$ inherently contains local noise from view inconsistencies, $k$-means provides crucial global smoothing over these probabilities.
>
> **Q4:** Our time efficiency does not come at the cost of hidden peak memory usage. CDAL's space complexity is strictly $\mathcal{O}(nd + nm + md)$, scaling linearly with sample size $n$. This is achieved through our formulation of the linearized HSIC gradient. By leveraging matrix associativity (e.g., computing $Z_u (\overline{G}^\top \overline{G})$ as $(Z_u \overline{G}^\top) \overline{G}$ in Eq. 16), we strictly avoid instantiating dense $n \times n$ similarity matrices.
>
> **Q5:** We agree that discussing theoretical boundaries strengthens the paper. First, if the non-Gaussianity assumption (Theorem 4.2) is violated, the model may only disentangle latent factors up to an arbitrary orthogonal rotation, potentially degrading fairness. Second, as the number of views $V$ increases, our equal weighting formulation might allow approximation errors from severely noisy or missing views to disproportionately corrupt the shared semantic graph $Z_u$.

---

> > ### Author Rebuttal · Reviewer_RBB9 · 2026-04-01
> >
> > The authors have address my concern.

---

### Official Review · Reviewer_3a9Q · 2026-03-11

**Soundness:** 3
**Presentation:** 3
**Significance:** 3
**Originality:** 3
**Overall Recommendation:** 5
**Confidence:** 4

**Summary:**

This paper introduces CDAL, a framework designed to tackle the fairness-utility trade-off and scalability bottlenecks in multi-view clustering. The authors frame the problem through a structural causal model, positing that observed data acts as a collider generated by entangled semantic and sensitive latent factors. To resolve this, they design a dual-anchor mechanism to reconstruct the data using separate semantic and sensitive anchor matrices. By enforcing strict structural orthogonality between these anchor sets and applying a linearized HSIC constraint on the semantic graph, the method isolates sensitive biases. An alternating optimization algorithm is provided, achieving linear time complexity. Empirical results across several benchmarks validate the approach's effectiveness.

**Compliance With Llm Reviewing Policy:**

Affirmed.

**Final Justification:**

The authors have addressed all my concerns, thus I maitain my positive score.

**Key Questions For Authors:**

Algorithm 1 initializes the semantic and sensitive anchors via k-means on the raw data. Given that the objective function is non-convex, how sensitive is the final disentanglement and clustering performance to this specific initialization? Have you observed different convergence behaviors when using random or strictly orthogonal initializations?

In Equation 6, the sensitive alignment utilizes a linear projection W. If the underlying sensitive attributes exhibit highly complex or non-linear structures, would a kernelized or non-linear alignment strategy be more appropriate, and would introducing such a strategy inevitably break the proposed linear complexity?

The current formulation simply sums the reconstruction errors across all views. In complex multi-view scenarios, views often contain varying degrees of noise. How might the CDAL framework be extended to incorporate view-specific weighting (e.g., adaptive view fusion) without compromising the structural orthogonality constraints?

**Limitations:**

Yes

**Strengths And Weaknesses:**

Strengths:
Formulating the fair multi-view clustering problem through a structural causal lens provides a highly interpretable and mathematically elegant motivation for the dual-anchor design.

The optimization strategy is highly efficient and well-crafted. Specifically, updating the anchors via the Sylvester equation and utilizing a linearized HSIC constraint with low-rank factorization ensures genuine linear scalability, which represents a substantial contribution to large-scale clustering methodologies.

Weaknesses:
The sensitive alignment term via a linear projection matrix W assumes a straightforward linear relationship between the sensitive graph Zs and the sensitive attributes G. This strict linearity might bottleneck the model's ability to capture and subsequently filter out complex, non-linear biases.

The objective function aggregates the multi-view reconstruction terms with equal weighting. This simple summation ignores the practical reality that different views often possess varying levels of quality, noise, or missing information, potentially allowing a severely degraded view to negatively impact the shared semantic graph Zu.

---

> ### Author Rebuttal · Authors · 2026-03-29
>
> **W1:** We agree a strictly linear projection $W^\top Z_s = G$ might be constrained for highly complex non-linear biases. Our primary motivation was to maintain strict $\mathcal{O}(n)$ complexity and ensure closed-form optimization. However, because $Z_s$ acts as a soft-assignment bipartite graph over local structural anchors $A_s$, it inherently preserves non-linear manifold information from the original feature space. Consequently, a linear projection on $Z_s$ is functionally equivalent to a non-linear projection on the raw data $X$.
>
> **W2:** We agree that practical views exhibit varying noise levels. Our deliberate choice of equal weighting ($\sum_{v=1}^V$) serves as a clean baseline to isolate and validate the proposed causal disentanglement mechanism without confounding performance gains with complex view-attention modules. However, extending CDAL to handle varying view qualities adaptively is highly feasible, as detailed in our response to Q3.
>
> **Q1:** Given the non-convex nature of our objective function, the initialization strategy plays a critical role in determining convergence behavior and the quality of local optima. To evaluate this, we compared our default $k$-means initialization with Random and Strictly Orthogonal (via QR decomposition) initializations in Table D.
>
> **Table D: Sensitivity analysis of different initialization strategies**
> | Initialization Strategy | Metric | Yale | Jaffe | COIL | Scene | Har | Zafar |
> | --- | --- | --- | --- | --- | --- | --- | --- |
> |  $k$-means (Ours) | ACC (%) | 60.36 | 96.24 | 71.17 | 37.65 | 75.26 | 98.88 |
> |  | Bal (%) | 12.50 | 33.33 | 75.00 | 83.04 | 14.29 | 68.64 |
> | Random Init | ACC (%) | 33.30 | 43.22 | 61.33 | 20.42 | 20.91 | 50.26 |
> |  | Bal (%) | 0.00 | 0.00 | 72.97 | 80.92 | 47.56 | 68.31 |
> | Orthogonal Init | ACC (%) | 34.52 | 42.91 | 62.04 | 20.54 | 20.73 | 50.26 |
> |  | Bal (%) | 7.69 | 0.00 | 64.29 | 82.28 | 41.67 | 68.31 |
>
> As shown in **Table D**, our default  $k$-means initialization consistently yields the highest clustering accuracy and robust fairness across all datasets. In contrast, both Random and Orthogonal initializations lead to severe utility degradation and frequently cause the fairness metrics to completely collapse. These empirical observations directly confirm that our  $k$-means initialization strategy is highly effective and essential for achieving the optimal fairness-utility trade-off.
>
> **Q2:** We agree that a kernelized or non-linear alignment strategy would be more appropriate for capturing highly complex, non-linear sensitive structures. The critical challenge in introducing such a strategy is avoiding the prohibitive $\mathcal{O}(n^2)$ computational cost typically associated with standard kernel matrices to preserve the scalability of our framework. Exploring scalable non-linear alignment mechanisms that can effectively filter out complex biases while strictly maintaining the overall $\mathcal{O}(n)$ complexity of our model is a highly valuable research direction.
>
> **Q3:** To address this, we have explored how the CDAL framework can be extended to incorporate adaptive view weighting without compromising the structural constraints. We can introduce learnable view weights $w_v$ (subject to $\sum_{v=1}^V w_v^\gamma = 1, w_v \ge 0, \gamma > 1$) to the multi-view reconstruction term:
>
> $$
> \min_{w_v, A, Z} \sum_{v=1}^V w_v^\gamma || X^{(v)} - (A_u^{(v)}Z_u + A_s^{(v)}Z_s) ||_F^2
> $$
>
> Crucially, the structural orthogonality constraint $\mathcal{L}_{orth}$ operates independently within the anchor spaces of each specific view. Therefore, $w_v^\gamma$ merely acts as a scalar multiplier to the gradients of $A_u^{(v)}$ and $A_s^{(v)}$ during alternating optimization. It automatically updates from noisy views without interfering with the orthogonality conditions.

---

> > ### Author Rebuttal · Reviewer_3a9Q · 2026-04-01
> >
> > The authors have address all my concern.

---

### Official Review · Reviewer_PCnT · 2026-03-12

**Soundness:** 3
**Presentation:** 3
**Significance:** 3
**Originality:** 3
**Overall Recommendation:** 5
**Confidence:** 5

**Summary:**

The paper introduces Causal Disentangled Anchor Learning (CDAL), a framework designed to address the fairness-utility trade-off and scalability bottlenecks in multi-view clustering. By framing the problem through a Structural Causal Model, the authors assume that the observed data is generated by entangled semantic and sensitive latent factors. To explicitly disentangle these, CDAL employs a dual-anchor mechanism that reconstructs data via separate semantic and sensitive anchor matrices. By enforcing strict structural orthogonality between these anchors and utilizing a linearized Hilbert-Schmidt Independence Criterion to ensure statistical independence, the method effectively isolates bias. The alternating optimization algorithm achieves an overall linear time complexity O(n), and empirical results demonstrate strong performance against various baselines.

**Compliance With Llm Reviewing Policy:**

Affirmed.

**Final Justification:**

The author’s detailed response has resolved my concerns, so I have increased my score from 4 to 5.

**Key Questions For Authors:**

Please refer to the weakness

**Limitations:**

The authors provided a brief societal impact statement, but the technical limitations section is somewhat limited. For instance, in Table 4 (Running time comparison), providing the specific hardware specifications (CPU/GPU models, RAM) used for these experiments would give a fairer context to the scalability claims.

**Strengths And Weaknesses:**

**strengths**:The alignment between the proposed Structural Causal Model and the actual optimization objectives (dual anchors + orthogonality constraint) is mathematically rigorous and highly interpretable.The derivation of the linearized HSIC is computationally clever. By leveraging low-rank factorizations and the associative property of matrix multiplication, the authors effectively bypass the $O(n^2)$ bottleneck of standard kernel matrices, making fair clustering truly scalable.**weaknesses**:While the quantitative results (ACC, NMI, Balance, MNCE) are solid, the paper lacks qualitative visualizations (such as t-SNE or PCA plots) of the learned latent representations $Z_u$. This leaves it unclear how well the semantic space is visually disentangled from the sensitive space.In Section 5.2, the hyperparameter search grid is provided, but the paper fails to explicitly state whether the optimal hyperparameters ($\alpha, \beta, \gamma$) are tuned individually per dataset or if a unified setting was found to work across all benchmarks.On the Zafar dataset (Table 2), baselines like SFD or KFC show a massive performance drop. The paper lacks a discussion on the specific characteristics of this dataset that cause such a drastic gap compared to CDAL.It is not specified whether the raw multi-view data requires strict normalization or standardization (e.g., z-score) prior to training to ensure that the structural orthogonality constraint works properly across views with potentially different
scales.

---

> ### Author Rebuttal · Authors · 2026-03-29
>
> **W1:** We agree that qualitative visualizations greatly enhance interpretability. We will include t-SNE visualizations of the learned latent representations in the experimental section of the final version.
>
> **W2:** To achieve the optimal performance reported in Table 2, hyperparameters ($\alpha, \beta, \gamma$) were tuned individually for each dataset using the provided grid. However, as illustrated in our sensitivity analysis (Fig. 7), CDAL demonstrates robust performance across a wide range. Clustering utility and fairness remain highly stable when $\alpha, \beta$, and $\gamma$ vary within $[10^{-1}, 10]$. This indicates that while per-dataset fine-tuning yields peak results, a unified default setting (e.g., $\alpha=1, \beta=1, \gamma=1$) serves as a strong baseline in practical unsupervised deployments.
>
> **W3:** The Zafar dataset is a synthetic benchmark specifically designed to exhibit extremely strong spurious correlations between semantic features and sensitive attributes. Traditional methods (e.g., SFD, KFC) employ soft correlation suppression or post-processing. In severely entangled datasets like Zafar, these methods indiscriminately discard semantic information to satisfy fairness constraints, causing a catastrophic drop in clustering accuracy (e.g., SFD's NMI drops to 2.00%). In contrast, CDAL tackles this structurally. Our dual-anchor mechanism explicitly isolates sensitive factors into a separate orthogonal subspace ($\mathcal{L}_{orth}$) during graph construction. This physical causal path blocking preserves the cluster-discriminative structure, maintaining a high NMI (91.31%) alongside competitive fairness.
>
> **W4:**
> Normalization is an essential preprocessing step in our experiment. Prior to training, we apply min-max normalization (scaling features to $[0, 1]$) to the raw data of each view. Since different modalities often possess vastly different numerical scales, omitting this step would cause the structural orthogonality constraint ($||{A_u^{(v)}}^\top A_s^{(v)}||_F^2$) and the reconstruction losses to be disproportionately dominated by views with larger numerical ranges. Normalization ensures that the dual-anchor subspaces are optimized equitably across all views, thereby enabling stable optimization and effective disentanglement. We will explicitly incorporate this preprocessing detail into Section 5.2 of the revised manuscript.
>
> **Limitations:**
> All running time experiments reported in Table 4 were conducted on a desktop computer equipped with an Intel Core i9-10900X CPU and 64GB of RAM. We will include this hardware context and explicitly expand the discussion on technical limitations in the final version.

---

> > ### Author Rebuttal · Reviewer_PCnT · 2026-04-03
> >
> > The author’s detailed response has resolved my concerns, so I have increased my score from 4 to 5.

---

### Official Review · Reviewer_Mk9h · 2026-03-12

**Soundness:** 3
**Presentation:** 3
**Significance:** 3
**Originality:** 2
**Overall Recommendation:** 4
**Confidence:** 4

**Summary:**

This paper proposes the Causal Disentangled Anchor Learning (CDAL) framework, which decouples latent representations into orthogonal semantic and sensitive subspaces via a dual-anchor mechanism. Integrating a linearized HSIC independence constraint and an alternating optimization algorithm, it ensures linear time complexity \(O(n)\) and the identifiability of disentangled factors while effectively balancing the utility and fairness of multi-view clustering.

**Compliance With Llm Reviewing Policy:**

Affirmed.

**Final Justification:**

The paper focuses on fair multi-view clustering via causal disentangled anchor learning, which targets the practical demand of unbiased representation learning in real-world multi-view data scenarios. The rebuttal effectively improves the theoretical clarity, empirical completeness and reproducibility of the paper.

**Key Questions For Authors:**

1) The linearized HSIC adopts a linear kernel, implicitly assuming a linear association between semantic representations and sensitive attributes. Have you tested the method’s performance in scenarios with non-linear correlations between the two? If the linear kernel fails, what alternative strategies do you propose?
2) The experiments use datasets with binary or simple categorical sensitive attributes. Have you validated the method on datasets with high-dimensional, hierarchical, or overlapping sensitive attributes? How does the dual-anchor mechanism adapt to complex sensitive factor structures?

**Limitations:**

Yes.

**Strengths And Weaknesses:**

Strengths:

1) From the perspective of structural causal models, it proposes a dual-anchor mechanism, which achieves physical separation of semantic and sensitive representations through orthogonality constraints.
2) By linearizing the HSIC independence constraint and leveraging the low-rank property of anchor graphs to avoid constructing n×n matrices, the algorithm’s time complexity is reduced to linear O(n).

Weaknesses:
1) The performance of the dual-anchor mechanism heavily relies on the number of anchors (m). As shown in the sensitivity analysis, excessive anchors introduce noise and redundancy that degrade clustering accuracy, while insufficient anchors may fail to capture complex semantic or sensitive subspaces.
2) This paper regards the orthogonality constraint ($L_{orth}$) between semantic anchors and sensitive anchors as a proxy for blocking the causal path from the sensitive factor to the semantic representation. However, this reasoning involves a logical leap. Structural orthogonality can only ensure the geometric separation of the two anchor subspaces, but it cannot prove that this separation corresponds to the actual path blocking in the causal relationship.
3) Applying the simplex constraint of sum equal to 1 and non-negative to the semantic graph $Z_u$ and the sensitive graph $Z_s$, and forcing them to be modeled in the form of a probability distribution, but this constraint lacks reasonable physical or semantic explanations:
a) The core of semantic representation is to capture the clustering-related features of the samples, rather than the probability distribution. The enforcement of non-negativity and normalization may limit the expressiveness of the representation.
b) This work did not verify the necessity of this constraint. If the constraint is removed, it may improve the clustering utility through more flexible representations without affecting fairness.
4) The authors claim that linear time complexity O(n), but this ignores the fact that the number of anchors (m) and sensitive attribute dimensions (h) may grow with data scale. The actual complexity O(nmh) can deviate significantly from linearity when m or h increases, making the claim misleading and overstating the method’s scalability for large-scale data.

---

> ### Author Rebuttal · Authors · 2026-03-29
>
> **W1:** We agree that the number of anchors $m$ controls the representational capacity of the latent subspaces. Small $m$ causes underfitting, excessively large $m$ captures local noise, degrading disentanglement. Empirically, setting $m \in [k, 5k]$ is a efficient heuristic capturing essential semantics while filtering instance-level noise, yielding stable performance (Fig. 4). We will add a discussion on selecting $m$ to guide deployment.
>
> **W2:** Structural orthogonality is a necessary but not sufficient condition for causal path blocking. True causal disentanglement relies on the joint effect of $\mathcal{L}_{orth}$ and statistical independence (linearized HSIC). Under our Linear SCM, geometric orthogonality physically ensures non-overlapping linear subspaces. Theorem 4.2 proves that when combined with statistical independence, true causal semantic factors are uniquely identifiable, severing spurious paths. We will refine Sec 3.2 to clarify orthogonality as the geometric foundation.
>
> **W3:** Physically, $Z_u \in \mathbb{R}^{m \times n}$ is a bipartite graph adjacency matrix between samples and anchors. Non-negativity and sum-to-one constraints ensure valid soft assignment distributions, essential for capturing the cluster-oriented topological structure for $k$-means. Without them, $Z_u$ loses interpretability as a graph. Empirically, removing these constraints degenerates the model into unconstrained matrix factorization. As **Table A** shows, this allows bias to leak back, causing a catastrophic loss of fairness (Balance drops to $0.00$ on most datasets) and severe utility degradation. This validates the constraints' necessity.
>
> **Table A: Ablation study on simplex constraints**
> | Variant | Metric | Yale | Jaffe | COIL | Scene | Har | Zafar |
> | --- | --- | --- | --- | --- | --- | --- | --- |
> | Ours | ACC (%) | 60.36 | 96.24 | 71.17 | 37.65 | 75.26 | 98.88 |
> |  | Bal (%) | 12.50 | 33.33 | 75.00 | 83.04 | 14.29 | 68.64 |
> | w/o Non-negative | ACC (%) | 70.72 | 54.09 | 63.91 | 20.20 | 46.59 | 78.94 |
> |  | Bal (%) | 0.00 | 0.00 | 42.86 | 80.61 | 0.00 | 0.00 |
> | w/o Sum-to-one | ACC (%) | 66.61 | 52.21 | 60.08 | 18.89 | 44.90 | 74.98 |
> |  | Bal (%) | 0.00 | 0.00 | 33.33 | 76.55 | 0.00 | 65.68 |
> | Unconstrained | ACC (%) | 70.46 | 55.07 | 60.31 | 18.80 | 46.60 | 78.94 |
> |  | Bal (%) | 0.00 | 0.00 | 42.86 | 80.00 | 0.00 | 0.00 |
>
> **W4:** In asymptotic time complexity analysis ($n \to \infty$), sensitive attribute dimension $h$ is a small constant. Similarly, $m$ is a hyperparameter independent of $n$ (typically $c \times k$). Because $m \ll n$ and $h \ll n$, they function as bounded constants. Thus, the cost $\mathcal{O}(nmh)$ grows strictly linearly w.r.t. sample size $n$. We will explicitly define $m$ and $h$ as bounded constants relative to $n$ in Sec 4.1.
>
> **Q1:** We chose the linear kernel ($K_{Z_u} = Z_u^\top Z_u$) for strict $\mathcal{O}(n)$ scalability. For complex non-linear associations, our framework integrates Random Fourier Features (RFF). By mapping $Z_u$ to a random feature space $\Phi(Z_u) = \sqrt{2/D} \cos(W_{rff}^\top Z_u + b_{rff}) \in \mathbb{R}^{D \times n}$ ($D > m$), we approximate an RBF kernel HSIC. Gradients are still efficiently computed via matrix multiplication, preserving $\mathcal{O}(nDh)$ complexity. As **Table B** shows, RFF-HSIC effectively handles non-linear scenarios. While improving accuracy slightly on some datasets, linear HSIC provides a highly stable fairness-utility trade-off across most benchmarks, confirming linear approximation suffices standardly, while RFF serves severe non-linear cases.
>
> **Table B: Comparison between linear HSIC and non-linear RFF-HSIC**
> | Variant | Metric | Yale | Jaffe | COIL | Scene | Har | Zafar |
> | --- | --- | --- | --- | --- | --- | --- | --- |
> | Linear HSIC | ACC (%) | 60.36 | 96.24 | 71.17 | 37.65 | 75.26 | 98.88 |
> |  | Bal (%) | 12.50 | 33.33 | 75.00 | 83.04 | 14.29 | 68.64 |
> | RFF-HSIC | ACC (%) | 69.67 | 83.85 | 63.19 | 38.39 | 76.02 | 99.62 |
> |  | Bal (%) | 11.11 | 33.33 | 60.00 | 77.67 | 12.86 | 65.21 |
>
> **Q2:** Our formulation natively supports overlapping attributes. To encode $p$ intersecting traits, we simply concatenate their one-hot representations into $G \in \{0,1\}^{hp \times n}$. The HSIC term and $\mathcal{L}_{sens}$ treat this expanded $G$ as a multi-target regression problem, allowing the sensitive anchors $A_s$ to span a broader subspace without increasing optimization complexity. Validated on the Creditcard dataset using Marriage and Gender simultaneously (**Table C**), CDAL effectively suppresses compounded bias and achieves state-of-the-art fairness and utility.
>
> **Table C: Performance on Creditcard dataset**
> | Metric | Fair-MVC | SFD | KFC | DFMVC | AKAN | IMVC | **Ours** |
> |:---|:---|:---|:---|:---|:---|:---|:---|
> | **ACC(%)** | 31.54 | 40.30 | 24.26 | 42.74 | 35.98 | 40.96 | **42.88** |
> | **Bal(%)** | 29.79 | 30.28 | 15.87 | 28.30 | 29.41 | 30.29 | **31.01** |

---

> > ### Author Rebuttal · Reviewer_Mk9h · 2026-04-02
> >
> > Thanks for the authors‘ response. My concerns have now been resolved.

---

### Decision · Program_Chairs · 2026-04-30

**Decision:**

Accept (regular)

**Comment:**

This paper presents a framework, referred to as causal disentangled anchor learning (CDAL) which employs a dual-anchor mechanism to
decouple latent representation into orthogonal semantic and sensitive subspaces. An additional linearized Hilbert-Schmidt Independence criterion (HSIC) constraint, CDAL ensures statistical independence from sensitive attributes while achieving linear time complexity $\mathcal{O}(n)$. All reviewers feel that such dual-anchor mechanism with the linearized HSIC constraint is a reasonable way to enforce the causal disentanglement. Most of concerns raised by reviewers were resolved during the rebuttal period, thanks to the authors' efforts for making most of them clarified. In particular, I like one response made by the authors, "Structural orthogonality is a necessary but not sufficient condition for causal path blocking. True causal disentanglement relies on the joint effect of and statistical independence (linearized HSIC). Under our Linear SCM, geometric orthogonality physically ensures non-overlapping linear subspaces. Theorem 4.2 proves that when combined with statistical independence, true causal semantic factors are uniquely identifiable, severing spurious paths." This clearly justifies the main contribution of this paper. The authors did a wonderful job in their response. Most of reviewers are now positive after the rebuttal.